# Stable intracranial imaging of dura mater-engrafted pancreatic islet cells in awake mice

Philip Tröster [1] ✉, Montse Visa [1], Ismael Valladolid-Acebes[1], Martin Köhler [1] & Per-Olof Berggren[1,2,3,4]

By transplanting pancreatic islets onto the dura mater of the mouse brain, we establish a microscopy platform that enables longitudinal intravital imaging of otherwise optically inaccessible tissue. The system combines a cranial window with an air-cushioned floating arena and stable head fixation, providing high mechanical stability for repeated single-cell $Ca^{2+}$ imaging sessions of up to 90 min in awake mice. We show that dura mater-engrafted islets integrate with host vascular and neural networks, and that human islet grafts secrete C-peptide in response to glucose stimulation, indicating metabolic integration. With this platform, we monitor anesthesia-induced changes in capillary blood flow and islet $Ca^{2+}$ dynamics. In awake mice, following subcutaneous glucose injection, we characterize intracellular $Ca^{2+}$ oscillations in insulin-secreting β-cells, revealing changes in amplitude, period, and plateau fraction while network coordination remains stable. The dura mater thus offers long-term optical access to functional endocrine tissue, facilitating stable intravital imaging under anesthesia-free, physiological conditions.

Intravital microscopy is one of the most genuine techniques for visualizing dynamic cellular and molecular processes in living organisms to better understand biological systems and pathological processes. However, microscopic observations in live animals are usually associated with obstacles, such as the need for optical access, the lack of repeatability of experiments, and the use of anesthesia. In this study, we present an imaging platform that combines intravital microscopy with tissue transplantation onto the dura mater – the outermost meningeal layer of the mouse brain – to enable long-term, minimally invasive, and stable monitoring of tissue grafts at single-cell resolution in vivo. An important advantage of this platform is its compatibility with confocal imaging in awake mice, eliminating anesthesia-related confounds. To enable this, we employed the Mobile HomeCage (MHC) technology[1], originally developed for imaging brain function in freely moving, awake rodents and increasingly adopted in neuroscience research[2]. A cranial window provides optical access to the cortical layers of the brain, while a surrounding metal mount enables highly stable fixation beneath the microscope objective. Building on this, we

identified the mouse dura mater as an accessible site for tissue transplantation. To demonstrate proof of concept, we transplanted isolated pancreatic islets onto the dura mater based on three key considerations. First, pancreatic islets are small, self-contained micro-organs[3] with well-documented engraftment capacity and preserved metabolic function at various transplantation sites, including the subcutaneous space[4], kidney capsule[5], liver[6], and the anterior chamber of the eye[7]. Second, the functional analysis of pancreatic islets has been extensively studied both in vitro[8] and in vivo[9], allowing us to contextualize our anesthesia-free and engraftment data with established frameworks. Third, we previously examined $Ca^{2+}$ dynamics in pancreatic islets engrafted in the anterior chamber of the eye either under isoflurane or Hypnorm (fentanyl, fluanisone, and midazolam) anesthesia[10]. Anesthetized animals exhibited marked alterations in islet $Ca^{2+}$ signaling and impaired glucose handling[10], highlighting the need for imaging approaches that preserve physiological conditions, such as those in awake animals. Understanding islet function under physiological conditions is particularly important given their central

[1]The Rolf Luft Research Center for Diabetes and Endocrinology, Karolinska Institutet, Stockholm, Sweden. [2]Diabetes Research Institute, University of Miami Miller School of Medicine, Miami, FL, USA. [3]West China Hospital, Sichuan University, Chengdu, China. [4]Tecnológico de Monterrey, School of Medicine and Health Sciences, Monterrey, NL, Mexico. ✉e-mail: philip.troster@ki.se

role in maintaining glucose homeostasis, making pancreatic islets a biologically and clinically relevant model. Pancreatic islets are micro-organs that form the endocrine pancreas and regulate glucose levels by secreting hormones such as insulin and glucagon. Pulsatile insulin secretion[11] is critical for optimal blood glucose regulation and is coordinated by a synchronized network of pancreatic β-cells[12], which is modulated by glucose stimulation[13], $Ca^{2+}$ signaling[14], gap junction-mediated electrical coupling[15], and external chemical signals such as hormones[16] and neurotransmitters[17]. Insulin secretion results from a signaling cascade involving glucose uptake and metabolism, ATP-induced closure of $K_{ATP}$ channels, and the opening of voltage-gated L-type $Ca^{2+}$ channels[18]. Oscillatory $Ca^{2+}$ influx into the cytoplasm of pancreatic β-cells mediates insulin release[18], with the resulting rise in cytoplasmic free $Ca^{2+}$ concentration ($[Ca^{2+}]_i$) triggering exocytosis of insulin-containing vesicles into the bloodstream[18]. In this study, we use $[Ca^{2+}]_i$ dynamics in pancreatic β-cells as a functional readout by expressing the genetically encoded calcium sensor GCaMP3, a well-established approach[19]. This method provides an indirect yet reliable measure of stimulus-secretion coupling, as oscillations in intracellular $Ca^{2+}$ closely mirror β-cell activity. We focus on the slow oscillatory pattern of β-cell $[Ca^{2+}]_i$ activity, transients with periods of 60–600 s, associated with coordinated insulin secretion[20]. Given that progressive disruption of β-cell $Ca^{2+}$ signaling is a hallmark of type 2 diabetes[21], in vivo monitoring of $[Ca^{2+}]_i$ dynamics holds significant clinical relevance. Beyond characterizing islet graft viability and engraftment processes such as vascularization and innervation, we show that human pancreatic islets xenotransplanted onto the dura mater retain hormone-releasing functionality and achieve metabolic integration with the host. We further establish subcutaneous glucose administration as a reliable method to stimulate elevated metabolic demand. Using the Mobile HomeCage system, we characterize $[Ca^{2+}]_i$ dynamics in awake mice following glucose stimulation and reveal how commonly used anesthetics modulate cellular $Ca^{2+}$ signaling. Collectively, our study establishes the dura mater as a transplantation site and long-term imaging platform for pancreatic islets. This approach enables high-stability, anesthesia-free intravital microscopy in awake mice, providing a physiologically relevant model for imaging tissues otherwise inaccessible to optical methods.

## Results

### Anesthesia-free intravital microscopy of pancreatic islets transplanted onto the dura mater

Pancreatic islets were isolated from transgenic donor mice on a C57BL/6J background expressing the genetically encoded calcium indicator GCaMP3 specifically in insulin-producing β-cells. After at least 1 week in culture, individual islets were transplanted onto the dura mater of syngeneic recipient mice via a cranial window. Each islet was placed directly onto the dura mater, the outermost meningeal layer of the brain, to promote vascular and neural integration into the host tissue (Fig. 1a). To ensure imaging stability, the cranial window was sealed with a glass coverslip and embedded in dental cement, and a metal headgear was mounted to enable rigid head fixation under the microscope (Fig. 1b). Using the Mobile HomeCage system, confocal microscopy was performed in fully awake, head-fixed mice, which were free to move within a low-friction, air-cushioned cage made of carbon fiber (Fig. 1c). After a 3-week integration period, animals underwent a 5-day habituation protocol to acclimate to the imaging setup. Awake-state recordings began at 4 weeks post-transplantation and continued for up to 6 months (Fig. 1d). Over the 5-day conditioning period, the initial stress-induced increase in blood glucose dropped to normal levels by day 5, indicating effective adaptation to the Mobile HomeCage environment (Supplementary Fig. 1a). To verify healthy engraftment and rule out post-surgical inflammation, we monitored systemic levels of TNF-α (tumor necrosis factor alpha), IFN-γ (interferon-gamma), and CRP (C-reactive protein) over a 4-week post-

transplantation period. No significant change of these pro-inflammatory markers was detected, and the cranial windows remained optically unobstructed, indicating that tissue engraftment proceeded without adverse immune reactions (Fig. 1e). The image stability achieved with confocal microscopy allowed clear identification of individual β-cells and consistent tracking of their GCaMP3-labeled cytosolic regions over time. Simultaneously, tissue reflectance (backscatter signal) was used to delineate the tissue outline and morphology of dura mater-engrafted pancreatic islets (Fig. 1f). Without motion correction or post-processing, the representative composite images and corresponding fluorescence traces from a 600-frame GCaMP3 recording at 1 frame per second (fps) demonstrate high spatiotemporal signal stability in awake, behaving animals (Fig. 1f). While the β-cell GCaMP3 signal exhibits characteristic $[Ca^{2+}]_i$ oscillations, the corresponding backscatter trace from the same region of interest remains stable with only minimal background noise, indicating no movement or signal drift (Fig. 1g).

### Vascularization and innervation of pancreatic islets transplanted onto the dura mater

Single confocal sections acquired 3 weeks after transplantation, following intravenous (i.v.) injection of the vascular tracer TMR-Dextran (2000 kDa), showed clear vascularization of the engrafted pancreatic islets. Capillary structures were distinguishable both within the GCaMP3 fluorescence signal, which marks the β-cell mass, and in the morphological backscatter signal, indicating full penetration of host vasculature into the graft tissue (Fig. 2a). A three-dimensional volumetric reconstruction, based on serial z-stacks acquired 5 weeks post-transplantation using lectin-649, confirmed a uniform distribution of blood vessels throughout the β-cell mass (Fig. 2b). To quantify the revascularization process, we analyzed serial z-stacks of TMR-Dextran-labeled islet vasculature and calculated vascular volume relative to the GCaMP3-labeled β-cell volume. Revascularization was first detected 1 week after transplantation, accounting for ~8% of the β-cell volume, and increased steadily to ~30% by week 10 (Fig. 2c). Absence of significant differences in vascular volume between weeks 4 and 10 suggests that revascularization is largely complete by week 4. During the angiogenic phase, the majority of newly formed intra-islet vessels exceeded 6 μm in diameter, with slightly thinner vessels observed early during graft integration (Fig. 2d). In addition to vascular integration, evidence of innervation provides further indication that dura mater-engrafted pancreatic islets are structurally incorporated into the host tissue. Cryosections taken 12 weeks post-transplantation revealed neural projections extending into the grafted tissue (Fig. 2e). Parasympathetic innervation was evidenced by punctate, often serially arranged fibers positive for the vesicular acetylcholine transporter (VAChT), distributed throughout the insulin-producing β-cell mass (Fig. 2f). Sympathetic innervation within the graft was limited to sparsely distributed tyrosine hydroxylase (TH)-positive cells (Fig. 2e). Although fiber-like sympathetic structures were not prominent within the islets themselves, TH-positive fibers were abundant in the underlying dura mater and extended toward the islet graft, indicating close anatomical proximity to sympathetic input (Fig. 2g).

### Vascular integrity and blood flow dynamics in dura mater-engrafted pancreatic islets

Addressing the dura mater as a previously uncharacterized site for islet transplantation, we investigated the vascular permeability properties of the de novo–formed islet graft capillary network by simultaneously i.v. injecting low- and high-molecular-weight fluorescently labeled dextrans during 15 min of intravital imaging. Mice were recorded 6 weeks after transplantation. We captured semi-volumetric dynamics of dextran extravasation from graft vasculature compared to surrounding dura mater vessels (Fig. 3a). Vascular permeability was quantified over time based on low molecular weight (10 kDa) dextran

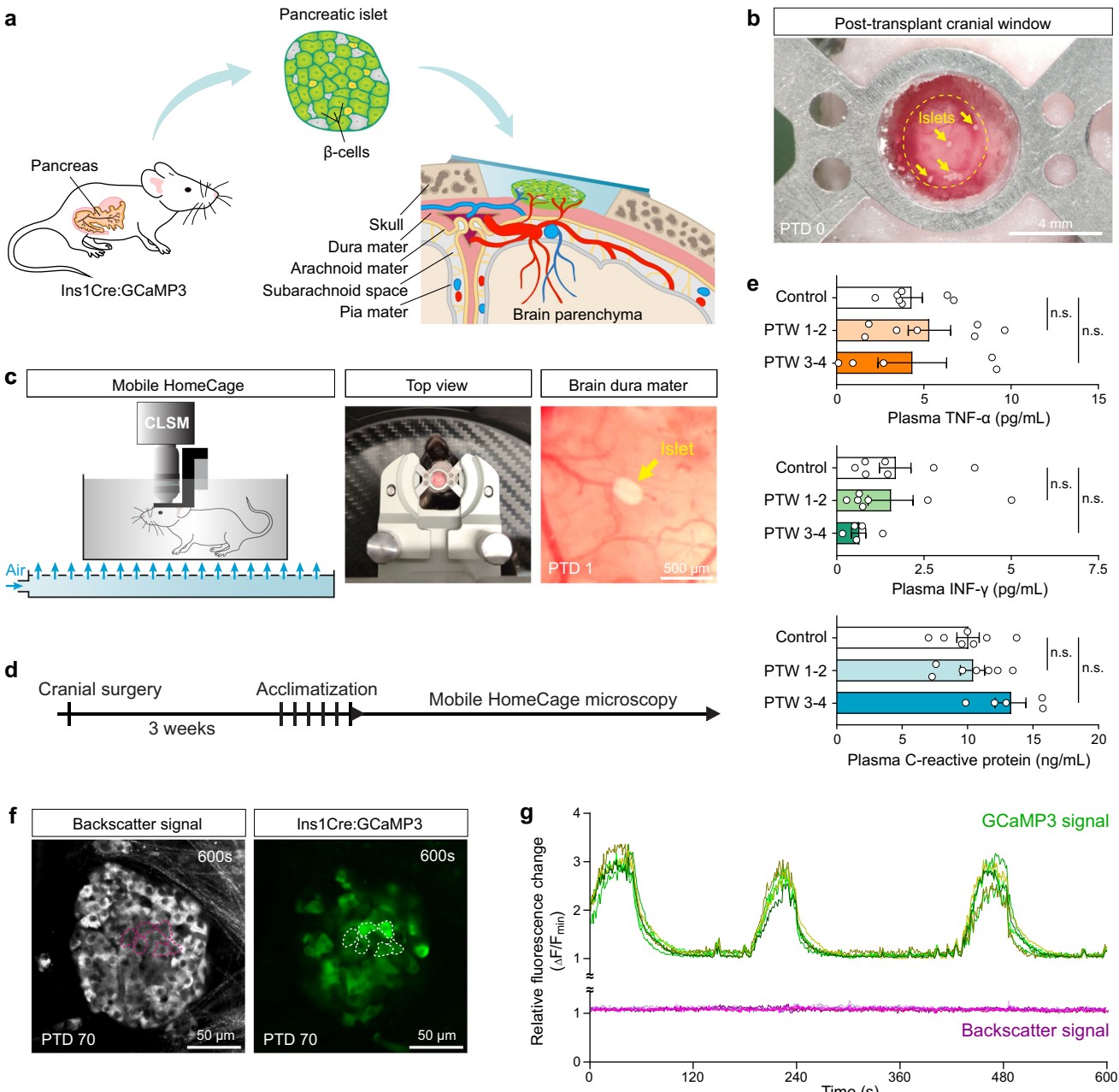

**Fig. 1 | Design and characterization of an anesthesia-free imaging platform for dura mater-engrafted pancreatic islets. a** Schematic of the experimental workflow. Pancreatic islets were isolated from donor mice expressing β-cell-specific GCaMP3 and transplanted onto the dura mater, followed by cranial window implantation. **b** Representative image of the craniotomy site on post-transplantation day 0 (PTD 0). Yellow dashed circle indicates the site of bone removal. Transplanted pancreatic islets (yellow arrows) are visible on the dura mater surface. The site is sealed with a glass coverslip and stabilized using a metal mounting ring embedded in dental cement. Representative of *n* = 18 mice. Scale bar, 4 mm. **c** Mobile HomeCage (MHC) setup for repeated confocal laser scanning microscopy (CLSM) in awake mice. Top view of the cranial mount secured in the microscope holder with the mouse inside the air-floating carbon fiber cage. Brain dura mater showing a transplanted islet on the dura at PTD 1. Representative of *n* = 18 mice. Scale bar, 0.5 mm. **d** Experimental timeline from transplantation to intravital imaging. Following a 3-week engraftment period, mice were habituated for 5 days and underwent imaging sessions (up to 90 min per session) conducted daily or weekly for up to 6 months. **e** Plasma levels of TNF-α, IFN-γ, and CRP in control and transplanted mice over 4 weeks. Data are mean ± s.e.m.; *n* = 7 (control), *n* = 7 (post-transplantation week 1-2), *n* = 5 (post-transplantation week 3–4). Statistical analysis, one-way ANOVA with Tukey's post hoc test. **f** Standard deviation z-projection of a 600-frame (1 fps) recording acquired 10 weeks post-transplantation in an awake mouse. Backscatter signal and GCaMP3 fluorescence show individual β-cells within a dura mater-engrafted islet (dashed magenta and white outlines). Scale bar, 50 μm. **g** Normalized fluorescence traces from the recording in (**f**), showing $[Ca^{2+}]_i$ dynamics and corresponding local backscatter signals from individual β-cells. Regions of interest correspond to outlines in (**f**). Signals are normalized to the minimum recorded value (ΔF/F$_{min}$).

extravasation (Fig. 3a). Islet graft vessels showed lower vascular permeability dynamics, with reduced extravascular accumulation of 10 kDa dextran normalized to the 2000 kDa reference dye (Fig. 3b). The difference in vascular integrity is substantiated by area-under-the-curve analysis (Fig. 3c) and a significantly reduced maximum permeability ratio in intra-islet capillaries compared to the adjacent vasculature (Fig. 3d). These results indicate that dura-engrafted pancreatic islets establish a tissue-specific, functionally integrated vasculature with greater stability and lower permeability than the surrounding dura mater. To further evaluate the physiological

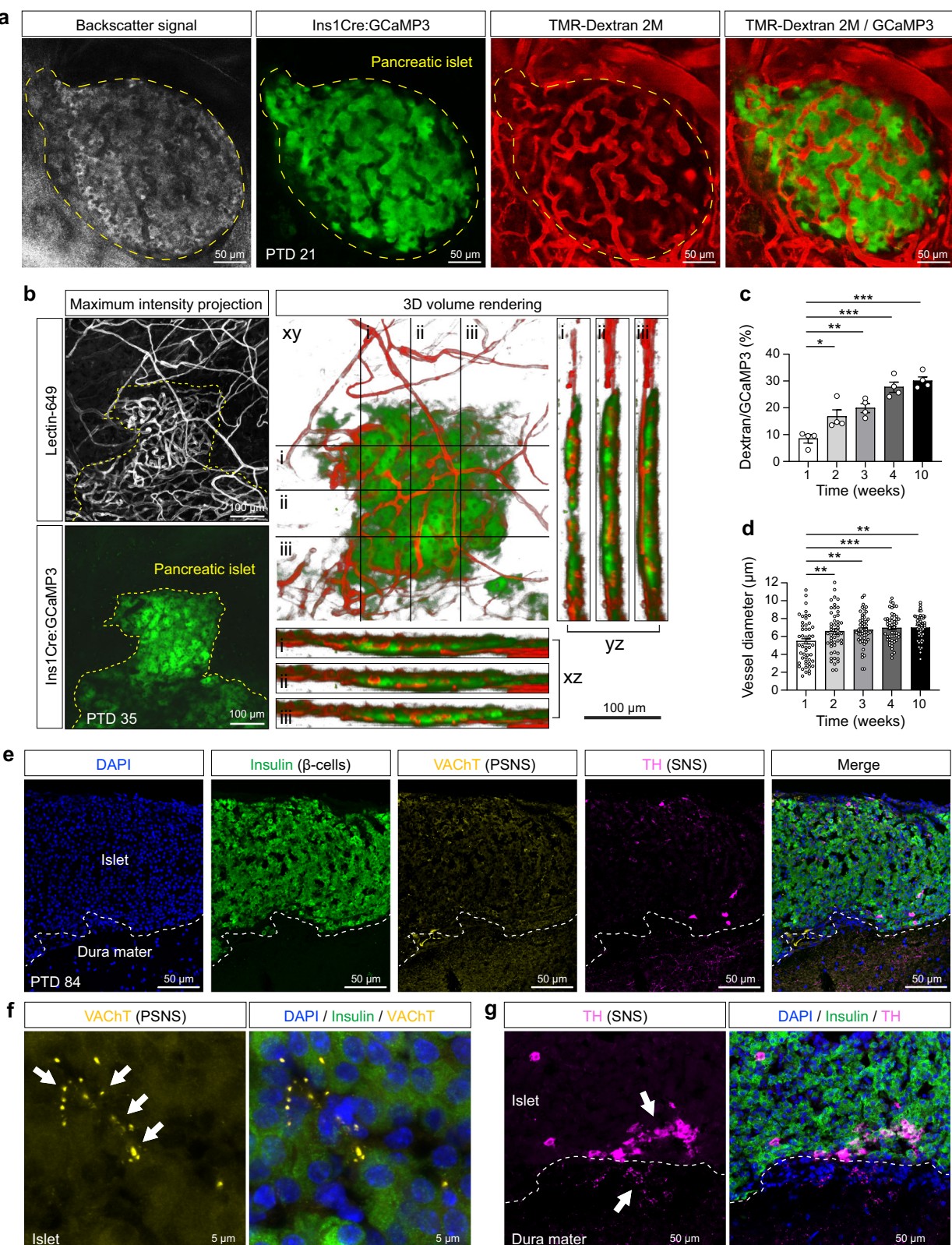

integration of dura mater-engrafted islets with the host vasculature, we quantified intra-islet blood flow velocity under both anesthetized and awake conditions, the latter enabled by intravital imaging in head-fixed, trained mice using the Mobile HomeCage system (Supplementary Movies 1–4). Measurements were performed sequentially in the same animal, first in the awake state and then following a brief rest period under isoflurane anesthesia. During intravital imaging,

blood vessels were visualized via intravenous injection of FITC-Dextran (2000 kDa), while islet morphology was delineated using tissue-specific backscattered reflection signals (Fig. 3e). Blood flow velocity was assessed by tracking fluorescently labeled red blood cells (RBCs) (Fig. 3e) and measuring the distance traveled per frame within the islet vascular plexus (Fig. 3f). A direct comparison in the same animal, first in the awake state and subsequently under isoflurane

**Fig. 2 | Revascularization and autonomic innervation of dura mater-engrafted pancreatic islets. a** Confocal optical section acquired 3 weeks post-transplantation showing a dura mater-engrafted islet (yellow dashed outline). Backscatter reflection delineates tissue structure, GCaMP3 fluorescence labels β-cells, and intravenously injected TMR-Dextran (2000 kDa) visualizes blood vessels. Representative of $n = 4$ mice. Scale bar, 50 μm. **b** Three-dimensional volumetric reconstruction of islet vasculature 5 weeks post-transplantation. Maximum intensity projection of a 42 μm z-stack shows GCaMP3-labeled β-cells and lectin-649-labeled vasculature. Lowercase roman numerals (i-iii) indicate positions of the corresponding xz and yz cross-sections. Scale bar, 100 μm. **c** Quantification of vascular volume (TMR-Dextran signal) relative to β-cell volume (GCaMP3 signal) at 1, 2, 3, 4, and 10 weeks post-transplantation, showing progressive revascularization. Data are mean ± s.e.m.; $n = 4$ islets. Statistical analysis, one-way ANOVA with Tukey's post hoc test; $P = 0.03$ (*), $P = 0.002$ (**), $P = < 0.001$ (***). **d** Capillary diameter measurements within the β-cell volume at the same time points. Each data point represents a vessel section (50 vessel segments from 4 islets per time point). Data are mean ± s.e.m.; $n = 4$ islets. Statistical analysis, one-way ANOVA with Tukey's post hoc test; $P = 0.008$ (**), $P = 0.003$ (**), $P = < 0.001$ (***), $P = 0.007$ (**). **e** Immunohistochemical analysis of a dura mater-engrafted pancreatic islet (15 μm cryosection) 12 weeks post-transplantation, showing autonomic innervation from sympathetic (SNS) and parasympathetic (PSNS) nervous systems. Nuclei are stained with DAPI (blue); β-cells with insulin (green); parasympathetic fibers with vesicular acetylcholine transporter (VAChT, yellow); sympathetic fibers with tyrosine hydroxylase (TH, magenta). White dashed lines mark the graft-host interface. Representative of $n = 2$ mice. Scale bar, 50 μm. **f** High-magnification view of autonomic fiber localization within the graft. VAChT-positive parasympathetic fibers (white arrows) co-stained with insulin (β-cells) and DAPI. Representative of $n = 2$ mice. Scale bar, 5 μm. **g** TH-positive cells (upper white arrow) and sympathetic fibers (lower white arrow) extending from the meningeal tissue into the islet graft. White dashed lines indicate the graft-host boundary. Representative of $n = 2$ mice. Scale bar, 50 μm.

anesthesia 1 h later, revealed a marked increase in blood flow velocity under anesthesia (Fig. 3g). Only a weak linear correlation was observed between RBC velocity and vessel diameter, consistent across both conditions. Under isoflurane anesthesia, the mean RBC velocity was increased by ~74% compared with the awake state (Fig. 3h). Vessel diameters remained unchanged between conditions, indicating that isoflurane did not affect vascular dilation in dura mater-engrafted islets (Fig. 3i).

## Pancreatic β-cell [Ca²⁺]ᵢ dynamics under anesthesia and in awake mice

The stability of our intravital microscopy platform enabled continuous monitoring of $[Ca^{2+}]_i$ dynamics over time at single-cell resolution in dura mater-engrafted islets expressing the β-cell-specific $Ca^{2+}$ indicator GCaMP3 (Fig. 4a). To isolate the effect of anesthesia under identical physiological conditions, we performed 1 h single-plane GCaMP3 imaging of a dura-mater-engrafted islet, capturing the transition from the awake state to isoflurane anesthesia and subsequent recovery in the same animal (Fig. 4b). The mouse was fully awake at the start and end of the recording, allowing us to delineate 5 distinct physiological states: awake, induction of anesthesia via breathing mask, deep sedation, recovery following cessation of isoflurane, and return to full wakefulness. The awake state was determined based on voluntary body movements and purposeful whisker activity displayed by the animal. The most prominent feature of the continuous $[Ca^{2+}]_i$ traces over the 1 h recording was a marked reduction in oscillation amplitude following the onset of isoflurane anesthesia (Fig. 4c). During the awake phase, GCaMP3 fluorescence exhibited robust, regular oscillations. Exposure to isoflurane initially caused a transient increase in amplitude, followed by a gradual decline. The onset of sedation was accompanied by an immediate decrease in amplitude and a concurrent lengthening of oscillation periods (Fig. 4d). In the fully anesthetized state, $[Ca^{2+}]_i$ activity was significantly altered, showing irregular and attenuated oscillations. Upon isoflurane withdrawal, $[Ca^{2+}]_i$ activity recovered rapidly, with amplitudes approaching baseline awake levels, although remaining statistically distinct (Fig. 4c). Oscillation periods also remained markedly prolonged (Fig. 4d). To assess the impact of isoflurane on β-cell network dynamics over the 1 h recording, we analyzed individual $[Ca^{2+}]_i$ traces from all recorded β-cells within a single focal plane of the dura mater-engrafted islet. Pairwise correlation analysis revealed that isoflurane modulated the network coherence, leading to diminished synchronization during sedation. Topographic mapping showed that this effect was spatially heterogeneous, with specific islet regions exhibiting more pronounced disruption of β-cell network activity than others (Fig. 4e).

To quantify anesthetic-induced alterations in pancreatic islet $[Ca^{2+}]_i$ dynamics, we performed repeated recordings in the same islet grafts sequentially under isoflurane anesthesia, Hypnorm sedation, and in the awake state (Supplementary Movies 5–7). Representative

$[Ca^{2+}]_i$ traces from each condition illustrate distinct β-cell activity patterns (Fig. 4f), forming the basis for subsequent quantitative analysis. Mean post-measurement blood glucose levels were 8.8 ± 0.5 mmol/L in awake mice, 8.4 ± 0.8 mmol/L under Hypnorm, and 12.3 ± 0.8 mmol/L under isoflurane anesthesia. Because β-cell $[Ca^{2+}]_i$ activity and islet metabolism are tightly coupled to circulating glucose, glucose monitoring is essential for data interpretation. As real-time glucose measurements during $[Ca^{2+}]_i$ imaging in awake animals were not feasible, glucose tolerance tests (GTTs) were performed in reference cohorts following subcutaneous (s.c.) glucose administration (2 g/kg body weight). Time-course analysis of blood glucose and insulin levels revealed normal glucose clearance and insulin secretion in awake and Hypnorm-sedated conditions, whereas isoflurane-anesthetized mice failed to mount an insulin response despite persistent hyperglycemia (Fig. 4g). Analysis of β-cell $[Ca^{2+}]_i$ dynamics focused on slow oscillatory events lasting 60–600 s. Averaged data showed that isoflurane anesthesia significantly reduced the oscillation amplitude by ~48% (Fig. 4h). Under Hypnorm sedation, amplitudes were intermediate between those of the awake and isoflurane groups, differing significantly only from the latter. Oscillation periodicity displayed high variability across experiments but showed no significant differences among the 3 conditions (Fig. 4h). Mean correlation analysis indicated that anesthesia had only a mild effect on β-cell synchronization, with no significant reduction across conditions (Fig. 4i). Further characterization of $[Ca^{2+}]_i$ activity revealed that isoflurane anesthesia reduced the plateau fraction by ~26% (Fig. 4i), whereas Hypnorm yielded intermediate values that did not differ significantly from either awake or isoflurane states.

## Glucose-stimulated insulin secretion and β-cell [Ca²⁺]ᵢ dynamics in dura mater-engrafted islets of awake mice

To assess the metabolic integrity and secretory function of islet grafts transplanted onto the dura mater, we employed a partial metabolic transplantation strategy combined with cross-species peptide differentiation. In this setup, 50 human pancreatic islets were engrafted onto the meningeal surface of immunodeficient Rag1⁻ᐟ⁻ mice (Fig. 5ai). The grafts remained viable, with vascular integration confirmed 4 weeks after transplantation (Fig. 5aii–iii). Tissue-specific light scattering facilitated identification of dura mater-engrafted human islets, while intravenous lectin-649 labeling delineated the surrounding host vasculature and islet capillary network. Detection of human C-peptide in serum, co-secreted with human insulin from the islet graft, enabled discrimination from endogenous murine β-cell secretion. The functional quality of the human islet preparations was verified prior to transplantation by assessing glucose-stimulated $Ca^{2+}$ responses and insulin secretion (Supplementary Fig. 1b, c). Static measurements of blood glucose, mouse insulin, and human C-peptide under fed and fasted conditions demonstrated functional metabolic integration of the human grafts,

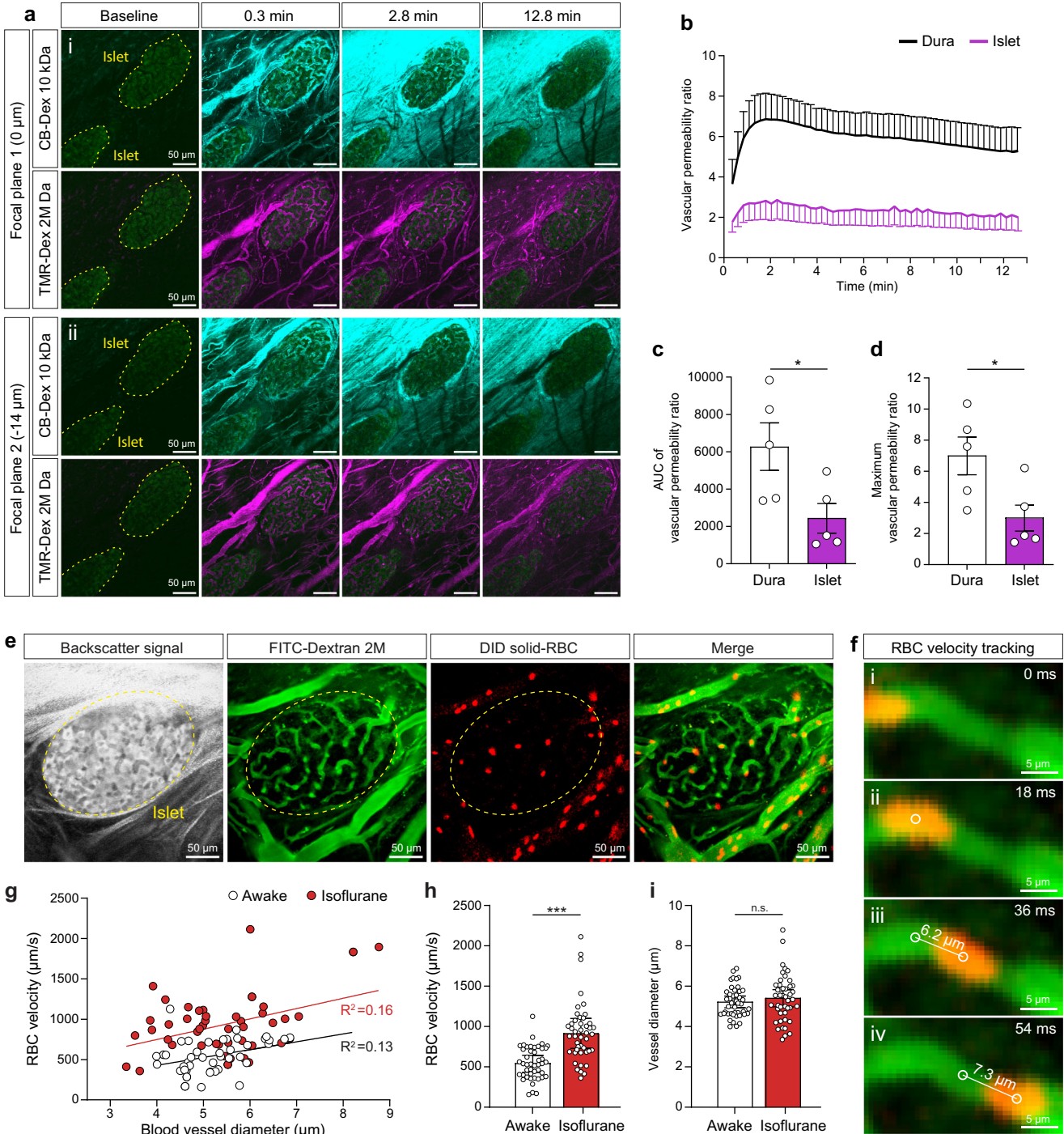

**Fig. 3 | Vascular integrity and blood flow dynamics in dura mater-engrafted pancreatic islets. a** Time-lapse confocal imaging of pancreatic islets on the dura mater (outlined by yellow dashed lines) 6 weeks post-transplantation. Vascular permeability was assessed using co-injected high- and low-molecular-weight dextrans: TMR-Dextran (2000 kDa, magenta) and CB-Dextran (10 kDa, cyan). Time 0 min marks the i.v. injection. Two focal planes (i, ii) were acquired every 14 s. Scale bar, 50 μm. **b** Vascular-permeability (VP) ratio, calculated as the fluorescence-intensity ratio of CB-Dextran (fast-leaking) to TMR-Dextran (slow-leaking) over 12.8 min post-injection. Baseline-subtracted values are shown for dural vessels (black) and islet capillaries (magenta). Data are mean ± s.e.m.; $n = 5$ islets. **c** Area under the curve (AUC) of the VP ratio for dura mater versus islet vasculature. Data are mean ± s.e.m.; $n = 5$ islets. Statistical analysis, paired two-sided $t$-test; $P = 0.035$ (*). **d** Maximum VP ratio observed in dura mater vessels versus islet capillaries. Data are mean ± s.e.m.; $n = 5$ islets. Statistical analysis, paired two-sided $t$-test; $P = 0.038$ (*). **e** Confocal images from red blood cell (RBC) velocity recordings 6 weeks post-transplantation. Backscatter reflection

highlights tissue structure, FITC-Dextran labels vasculature, and DiD labels RBCs. The merged image shows DiD-labeled RBCs within both intra-islet and dura mater vessels (islet outlined by yellow dashed line). Representative of $n = 2$ mice. Scale bar, 50 μm. **f** Consecutive high-speed fluorescence frames (i-iv, 55.5 fps) from an awake mouse showing a single DiD-labeled RBC traversing an islet capillary. Fluorescence-centroid tracking was used to determine RBC velocity. Scale bar, 5 μm. **g** Scatter plot of intra-islet RBC velocity as a function of vessel diameter in the same mouse imaged awake (white) and subsequently under isoflurane anesthesia (red). Each point represents one vessel segment (20 measurements). Linear-regression lines and corresponding $R^2$ values indicate correlation strength. **h** Mean RBC velocity in islet capillaries under awake and anesthetized conditions. Data are mean ± s.e.m.; $n = 45$ vessel segments. Statistical analysis, unpaired two-sided $t$-test; $P = < 0.001$ (***). **i** Corresponding vessel diameters during RBC velocity measurements under awake and anesthetized conditions. Data are mean ± s.e.m.; $n = 45$ vessel segments. Statistical analysis, unpaired two-sided $t$-test.

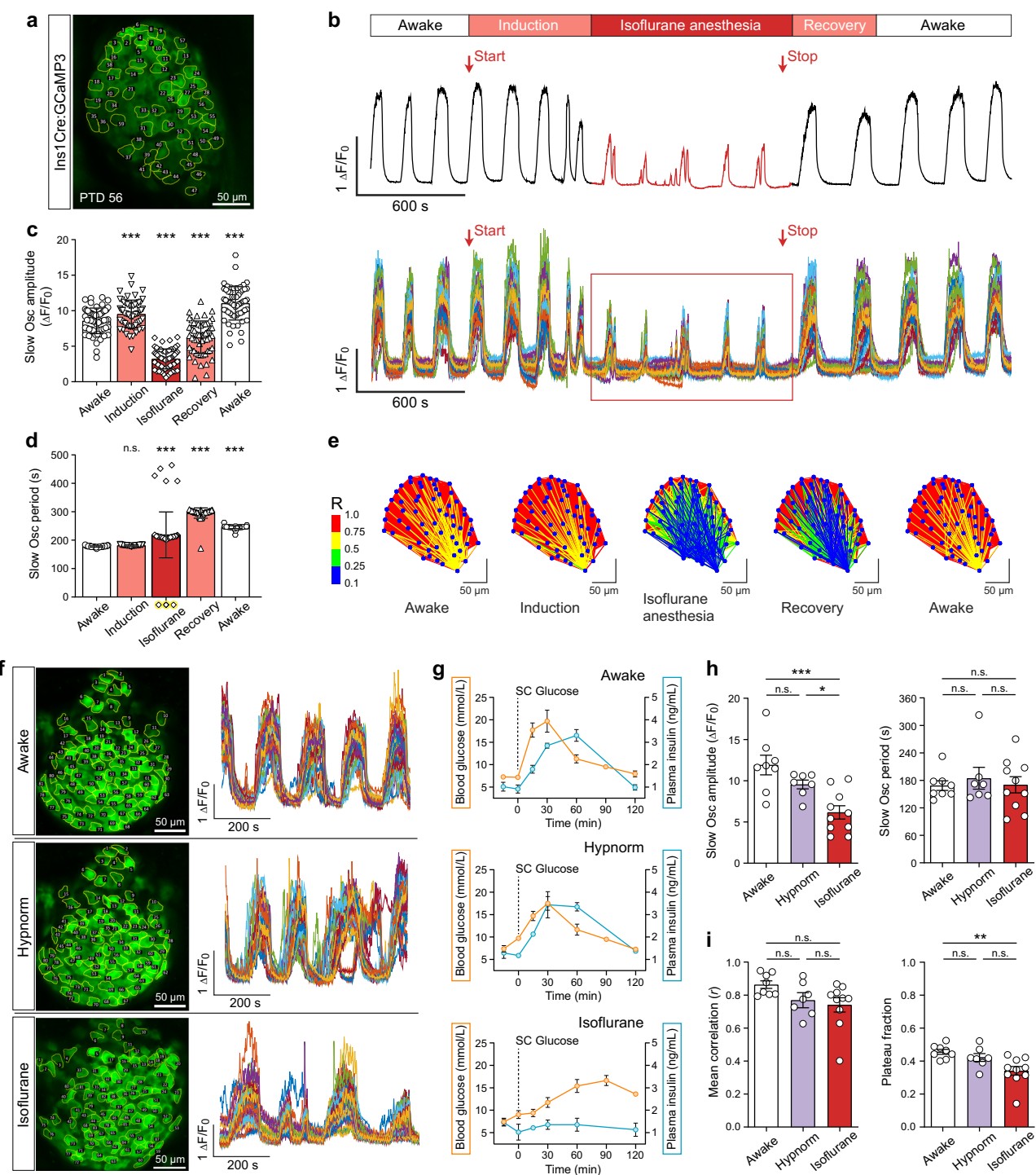

even in the absence of acute glycemic stimulation (Fig. 5b). Blood glucose, mouse insulin and human C-peptide levels exhibited physiologically appropriate elevations in the fed state and declined in parallel after a 6 h fasting period, consistent with glucose-regulated β-cell activity (Fig. 5b). Intraperitoneal glucose tolerance tests (IPGTTs) in transplanted host animals revealed a typical insulin secretory response, with plasma insulin peaking 15 min after glucose administration (Fig. 5c). Circulating human C-peptide levels increased in parallel, reaching a maximum at 30 min post-injection (Fig. 5c). These dynamics indicate that human islet grafts preserved their intrinsic glycemic set point for insulin release, exhibiting a delayed but distinct rise in C-peptide relative to the earlier response of host-derived mouse insulin (Fig. 5c).

Following confirmation of the metabolic integration of dura mater-engrafted islets into host physiology, we employed our microscopy platform to examine β-cell $[Ca^{2+}]_i$ dynamics in awake mice under glucose stimulation and varying metabolic demand. Subcutaneous injection of glucose into the lateral lumbar region was identified as a reliable and reproducible delivery route compatible with β-cell $[Ca^{2+}]_i$ imaging in awake, head-fixed mice within the Mobile HomeCage environment. To contextualize the recorded β-cell $[Ca^{2+}]_i$ responses, we first characterized glucose delivery kinetics to dura mater-engrafted islets and evaluated systemic glucose handling following s.c. administration. Using the fluorescent glucose analog 2-NBDG, we monitored glucose distribution to the dura mater in awake mice (Supplementary Movie 8). 2-NBDG fluorescence reached the dura

**Fig. 4 | Pancreatic β-cell [Ca²⁺]ᵢ dynamics and network activity in awake and anesthetized mice. a** Standard deviation z-projection from a 1 h GCaMP3 fluorescence recording (1 fps) of a dura mater-engrafted islet 8 weeks post-transplantation under transient isoflurane anesthesia. Fifty-nine individual β-cells used for [Ca²⁺]ᵢ trace analysis are outlined in yellow. Representative of $n = 3$ mice. Scale bar, 50 μm. **b** Baseline-normalized β-cell [Ca²⁺]ᵢ activity recorded from the focal plane shown in (a) during a full anesthesia cycle. Red arrows indicate the start and end of 2% isoflurane administration. Distinct physiological states – awake, induction, deep anesthesia, recovery, and awake – are indicated above the traces. Mean trace (upper panel) and single-cell traces (lower panel) are shown over a 60 min recording; $n = 59$ β-cells. **c** Mean slow [Ca²⁺]ᵢ oscillation amplitudes across the five anesthesia and wakefulness sub-periods. Individual β-cell values are overlaid as data points. Data are mean ± s.e.m.; $n = 59$ β-cells. Statistical analysis, two-way ANOVA with Bonferroni post hoc test; $P = < 0.001$ (***). **d** Mean slow [Ca²⁺]ᵢ oscillation periods across the same sub-periods. Yellow-circled data points indicate β-cells with out-of-range signals. Data are mean ± s.e.m.; $n = 59$ β-cells. Statistical analysis, two-way ANOVA with Bonferroni post hoc test; $P = < 0.001$ (***). **e** Topographic map

of β-cells showing pairwise correlation strength of [Ca²⁺]ᵢ activity during the transient isoflurane anesthesia shown in (**b**). Correlation strength is represented by the color-coded scale; $n = 59$ β-cells. **f** Representative Ca²⁺ imaging frames and corresponding single β-cell [Ca²⁺]ᵢ traces from dura mater–engrafted GCaMP3 islets recorded in awake mice, under Hypnorm anesthesia, and under isoflurane anesthesia. Scale bar, 50 μm. **g** Blood glucose and plasma insulin levels following s.c. glucose injection (2 g/kg body weight) under awake conditions, Hypnorm anesthesia, and isoflurane anesthesia. Data are mean ± s.e.m.; $n = 4$ (awake), $n = 3$ (Hypnorm), $n = 3$ (isoflurane). **h** Mean slow [Ca²⁺]ᵢ oscillation amplitudes and periods from individual Ca²⁺ imaging recordings across the three physiological states. Data are mean ± s.e.m.; $n = 8$ (awake), $n = 7$ (Hypnorm), $n = 10$ (isoflurane). Statistical analysis, one-way ANOVA with Tukey's post hoc test; $P = 0.015$ (*), $P = < 0.001$ (***). **i** Network correlation and plateau fraction of β-cell [Ca²⁺]ᵢ activity across the three physiological states. Data are mean ± s.e.m.; $n = 8$ (awake), $n = 7$ (Hypnorm), $n = 10$ (isoflurane). Statistical analysis, one-way ANOVA with Tukey's post hoc test; $P = 0.0044$ (**).

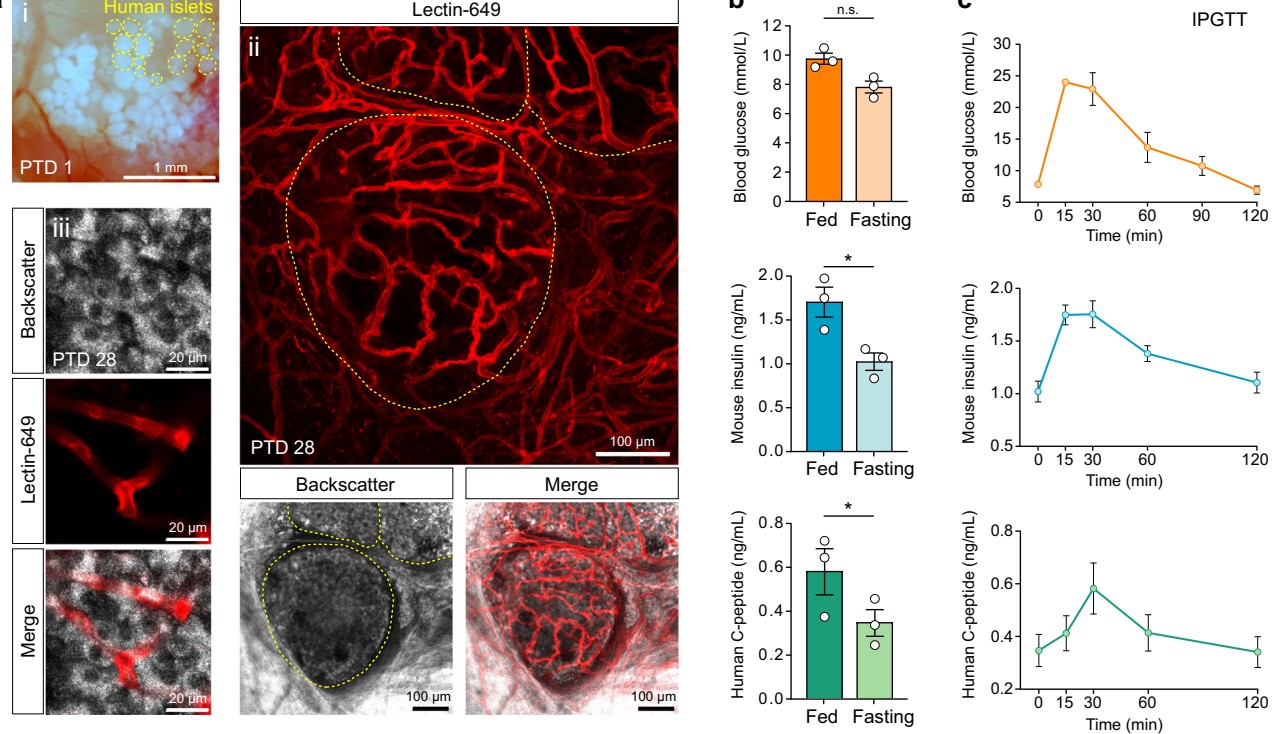

**Fig. 5 | Metabolic integration and C-peptide secretion from dura mater-engrafted human pancreatic islets. a** (i) Image of 50 human pancreatic islets one day after transplantation onto the dura mater. Yellow circles highlight representative single islet grafts. Scale bar, 1 mm. (ii) Vascularization of dura mater-engrafted human islets 4 weeks post-transplantation. The backscatter signal delineates islet morphology (yellow dashed line), while lectin-649 labeling visualizes the adjacent vasculature and islet graft capillaries in a multi z-stack projection. Scale bar, 100 μm. (iii) Single focal plane depicting intra-islet blood vessels. Scale

bar, 20 μm. Representative of $n = 2$ mice. **b** Static measurements of host blood glucose, endogenous mouse insulin, and human C-peptide under fed and 6 h fasted conditions. Data are mean ± s.e.m.; $n = 3$ independent experiments. Statistical analysis, paired two-sided t-test; $P = 0.09$ (n.s.), $P = 0.05$ (*), $P = 0.01$ (*). **c** Time course of host blood glucose, mouse insulin, and human C-peptide levels following i.p. glucose injection (2 g/kg body weight), measured over 120 min. Data are mean ± s.e.m.; $n = 3$ independent experiments.

mater and the peri-graft region within minutes, with half-maximal intensity detected at ~5 min and peak signal at ~17 min post-injection (Fig. 6a). Systemic glucose metabolism was evaluated in the same cohort of mice subsequently used for [Ca²⁺]ᵢ imaging. These mice carried β-cell-specific GCaMP3-expressing islets engrafted onto the dura mater for at least 8 weeks (Fig. 6b). Blood glucose and plasma insulin levels were measured over 60 min following the glucose challenge to define the β-cell secretory demand (Fig. 6c). These findings indicate that dura mater-engrafted islets are subjected to a progressively increasing insulin demand, which reaches a plateau between 15 and 60 min post-injection, delineating a physiologically relevant

window for functional imaging. Representative averaged single β-cell [Ca²⁺]ᵢ traces, recorded at 1 frame per second over 70 min, illustrate the final [Ca²⁺]ᵢ imaging experiments in awake, fasted mice. Subcutaneous glucose (2 g/kg body weight) was administered at 10 min, as indicated by the orange dashed line (Fig. 6d). Blood glucose concentrations measured before (7.5 ± 0.3 mM) and after (14.4 ± 0.8 mM) the recording confirmed effective glucose delivery. For quantitative analysis, the imaging period was divided into four temporal segments: a basal pre-stimulatory phase followed by three consecutive 20-min intervals post-injection (G 1–20, G 20–40, and G 40–60). Analysis of independent recordings revealed progressive decline in the amplitude and

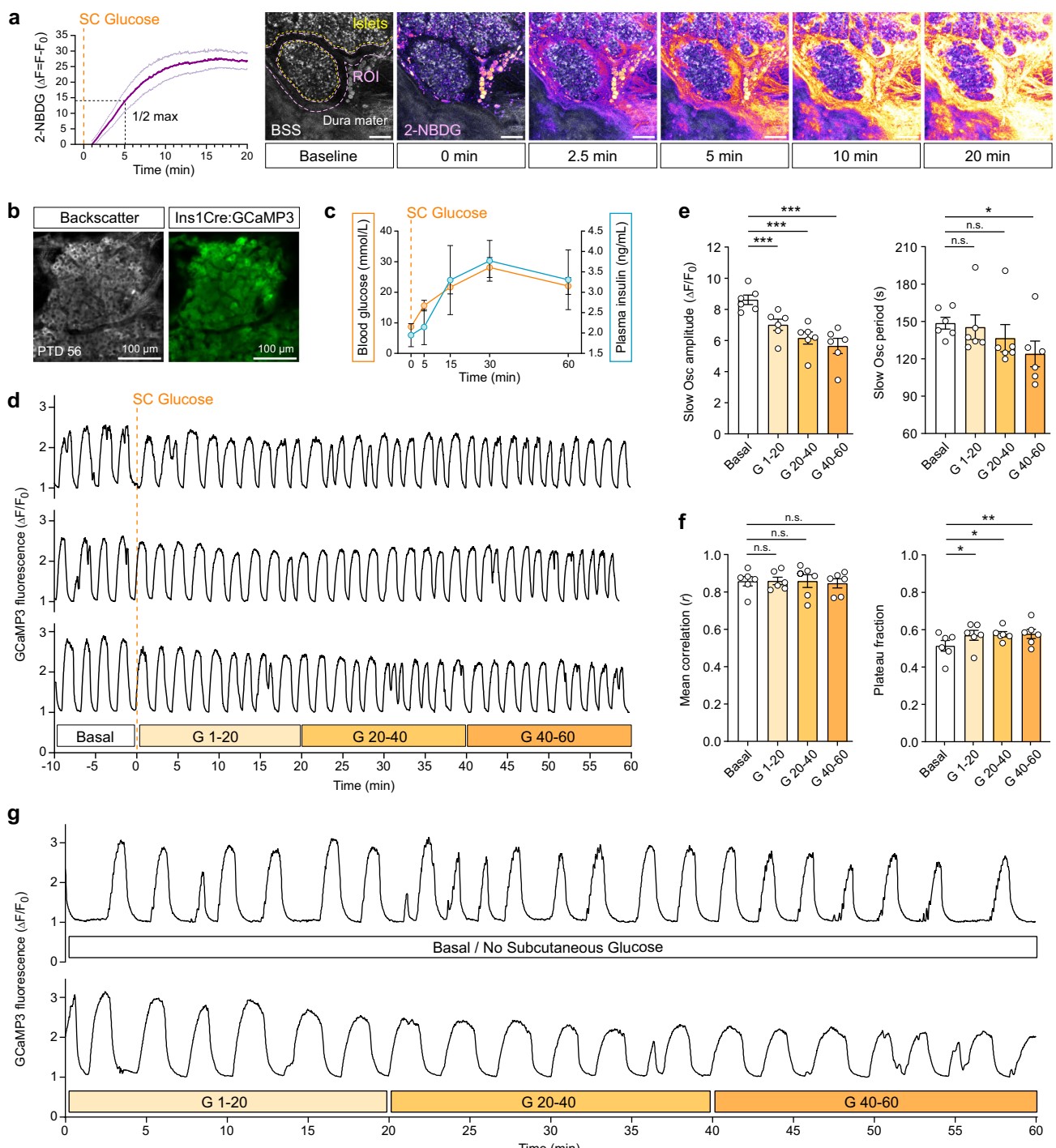

**Fig. 6 | Glucose-stimulated β-cell [Ca²⁺]ᵢ dynamics in dura mater-engrafted pancreatic islets of awake mice. a** Glucose delivery kinetics to dura mater-engrafted islets after subcutaneous (SC) injection. Co-injected fluorescent glucose analog 2-NBDG was imaged for 20 min at 1 fps (injection marked by orange dashed line). Baseline-subtracted mean fluorescence trace showing time to half-maximal and plateau uptake; *n* = 3 independent experiments. Representative baseline frame showing backscatter signal (BSS) with region of interest (ROI) and islet grafts (yellow dashed outline). Representative frames illustrate progressive 2-NBDG accumulation up to 20 min post-injection. Scale bar, 50 μm. **b** Representative focal plane of a dura mater−engrafted islet 56 days post-transplantation showing BSS and GCaMP3 fluorescence, demonstrating β-cell resolution for [Ca²⁺]ᵢ imaging; *n* = 3 mice. Scale bar, 100 μm. **c** Blood glucose and plasma insulin levels measured in awake mice over 60 min following s.c. glucose injection (2 g/kg body weight; orange dashed line). Data are mean ± s.e.m.; *n* = 3 mice. **d** Mean [Ca²⁺]ᵢ traces from ~60 β-cells in

awake mice, including 10 min of fasted baseline followed by 60 min post-glucose injection (orange dashed line). Defined analysis windows: baseline, glucose-stimulated 1–20 min (G 1–20), 20–40 min (G 20–40), and 40−60 min (G 40-60). Fluorescence normalized to baseline (ΔF/F₀); *n* = 3 independent experiments. **e** Slow [Ca²⁺]ᵢ oscillation amplitude and period across glucose stimulation windows. Data are mean ± s.e.m.; *n* = 6 independent experiments. Statistical analysis, two-way ANOVA with Bonferroni post hoc test; *P* = < 0.001 (***), *P* = 0.0127 (*). **f** β-cell synchronization (correlation coefficient) and plateau fraction across glucose stimulation windows. Data are mean ± s.e.m.; *n* = 6 independent experiments. Statistical analysis, two-way ANOVA with Bonferroni post hoc test; *P* = 0.0236 (*), *P* = 0.0124 (*), *P* = 0.0098 (**). **g** Comparison of mean β-cell [Ca²⁺]ᵢ traces over 60 min from the same animal under identical imaging conditions on two separate days, three days apart, with and without s.c. glucose (2 g/kg body weight) stimulation. A total of 140 individual β-cells were analyzed for each condition.

periodicity of slow $[Ca^{2+}]_i$ oscillations following glucose administration (Fig. 6e). Relative to baseline, oscillation amplitudes decreased by 19%, 29%, and 34% during G 1–20, G 20–40, and G 40–60, respectively. Inter-event intervals were also reduced, reflecting gradual increase in oscillation frequency of 2%, 8%, and 17% compared to pre-injection dynamics. Although β-cell synchronization remained largely unchanged, the plateau fraction remained elevated under sustained glucose stimulation (Fig. 6f). Compared to baseline, plateau fractions rose by 11%, 12%, and 12% across the respective intervals, indicating prolonged active $[Ca^{2+}]_i$ states in response to heightened metabolic demand. To exclude potential phototoxic effects from extended laser exposure, we performed control recordings in the same animal on separate days under identical imaging conditions. Each session comprised a 60 min recording, capturing either basal (non-stimulated) or post-glucose-injection (stimulated) islet $[Ca^{2+}]_i$ activity (Fig. 6g; Supplementary Movies 9, 10). The internal control confirmed that the attenuation in $[Ca^{2+}]_i$ amplitude reflected physiological glucose responses, not imaging artifacts. Under basal conditions, β-cell $[Ca^{2+}]_i$ traces displayed longer low-signal durations, whereas glucose stimulation markedly increased the time spent in the plateau phase. Elevated blood glucose levels modulated the dynamics of β-cell $[Ca^{2+}]_i$ oscillations by reducing amplitude, shortening quiescent intervals, and increasing plateau occupancy.

## Discussion

### The dura mater: a promising site for islet transplantation

In this study, we established the dura mater, the outermost meningeal layer of the mouse brain, as a stable and accessible site for the transplantation and long-term intravital imaging of micro-organs, such as isolated pancreatic islets. Combined with Mobile HomeCage microscopy, this approach enabled minimally invasive, longitudinal monitoring of islet physiology and intracellular β-cell $Ca^{2+}$ dynamics in awake mice, providing a robust platform for studying graft function under physiological conditions. Cranial window surgery for cortical imaging is a well-established procedure in experimental mouse models, offering optical access to the brain. Standardized protocols describe this technique[22] and provide guidance on surgical setup and troubleshooting[23]. To enable successful transplantation, several technical challenges were addressed, including prevention of graft compression due to brain bulging, maintenance of stable graft positioning, minimization of immune responses, and support of the graft during the initial hypoxic phase preceding vascular integration. Brain bulging was effectively prevented by limiting the craniotomy to a maximum diameter of 5 millimeter[24]. Graft stability was enhanced using 2.5% gelatin in Ringer's solution, which facilitated islet positioning on the dura mater. While grafts remained largely intact, occasional displacement of islets near the bone edge was observed during the first few days post-transplantation. Immune responses were minimized through syngeneic transplantation and refined surgical techniques. No antibiotic treatment was required, cranial windows remained optically clear, and inflammatory markers, including TNF-α, IFN-γ, and CRP, showed minimal or transient elevation. This indicates preserved immune stability at the graft site, consistent with previous reports following cranial window implantation[23]. The brain is a highly oxygenated organ, and studies have confirmed high capillary hemoglobin saturation in the somatosensory cortex, close to the region used for graft placement onto the dura mater[25]. However, prior to full vascular integration, transplanted tissue may experience transient hypoxia, potentially leading to hypoxia-induced cell death[26]. To mitigate this, cerebrospinal fluid was allowed to leak through a small puncture in the meninges, promoting passive diffusion of oxygen and nutrients to the islets during early engraftment. Successful engraftment was characterized by rapid vascularization arising from the meningeal circulation. As pancreatic islets were cultured for at least 1 week before transplantation, resulting in complete loss of donor-derived

endothelial cells[27], the newly formed vasculature likely originated entirely from host meningeal vessels. Approximately 50% vascularization was achieved by 2 weeks, with near-complete revascularization by 4 weeks, establishing robust vascular connectivity that ensures oxygen and nutrient delivery to sustain long-term graft function. These findings align with previous studies demonstrating the brain as a viable site for islet transplantation. In a rat dementia model, islets transplanted into the subarachnoid space restored insulin-dependent brain functions, confirming both viability and endocrine activity[28]. Islets engrafted on the dura mater revascularize similarly to those at established sites such as the anterior chamber of the eye[29], subcutaneous space[30], and kidney capsule[31], forming functional microvasculature within 2–4 weeks. The resulting capillary network resembled native islet architecture, with vessel diameters of 5 to 10 micrometers[32]. Quantitative analysis of capillary permeability within the graft and adjacent meningeal vasculature 6 weeks post-transplantation revealed that the newly formed blood vessels acquire distinct functional properties, likely influenced by cues from the islet microenvironment. This observation is consistent with previous findings showing that vascular endothelial growth factor A (VEGF-A), secreted by pancreatic islets, plays a key role in guiding the development and specialization of intra-islet vasculature[33]. Islet capillaries possess numerous fenestrations, transcellular pores that enable efficient molecular exchange across endothelial cells[34]. Dural blood vessels also display fenestrations but, unlike parenchymal vessels, lack tight junctions, resulting in markedly increased permeability[35]. In contrast, pial vessels within the subarachnoid space contribute to cerebral perfusion and form part of the blood-brain barrier (BBB) through restrictive tight junctions[36]. The permeability pattern observed at 6 weeks post-transplantation indicates that intra-islet vessels exhibit tighter permeability control than surrounding dural vessels, consistent with engraftment occurring outside the BBB. These findings highlight the capacity of the dura mater to support effective islet revascularization and underscore the influence of the graft microenvironment on angiogenesis.

Islet graft integration was accompanied by reinnervation through the autonomic nervous system. The dura mater overlying the somatosensory cortex is primarily innervated by sensory fibers of the trigeminal nerve[37], with additional sympathetic input from the superior cervical ganglion[38] and parasympathetic fibers from the sphenopalatine ganglion[39]. In contrast, native pancreatic islets receive sympathetic innervation from the coeliac ganglion and parasympathetic input from the vagus nerve[40]. Despite these differing origins, dura mater-engrafted islets developed parasympathetic cholinergic innervation, with acetylcholine, the key neurotransmitter, known to regulate islet metabolic activity[41]. Sympathetic reinnervation was also detected but less pronounced, appearing primarily as tyrosine hydroxylase-positive cellular structures. Overall, the innervation pattern resembled that of intraocular islet transplants[41], which closely recapitulate native islet innervation[41]. Although functional validation remains pending, the observed reinnervation likely supports physiological integration and contributes to graft functionality.

### Physiological divergence between anesthetized and awake states

As a determinant of endocrine islet function, blood flow velocity measurements in awake mice confirmed successful graft integration and supported the viability of the dura mater as a transplantation site. Adequate perfusion ensures efficient delivery of oxygen, nutrients, and glucose while facilitating waste clearance and hormone dispersion. Consistent with our findings, intra-islet capillaries in the native pancreas exhibit RBC velocities of up to 500 μm/s, depending on vessel segment and flow direction[42]. In our model, isoflurane anesthesia significantly increased intra-islet RBC velocity, suggesting a potential confounding effect during intravital imaging. This increase may partly

reflect isoflurane-induced hyperglycemia[43], as islet blood flow is highly glucose-sensitive, rising markedly between hypoglycemic (2.8 mM) and hyperglycemic (16.8 mM) states[44]. Pericyte-mediated regulation of capillary diameter, driven by glucose-dependent adenosine signaling from ATP co-released with insulin, contributes to this effect[45]. However, the suppression of insulin secretion by isoflurane and the absence of overt vessel dilation suggest that this mechanism plays a limited role. A more likely explanation is that the meningeal vasculature supplying dura mater-engrafted islets is particularly responsive to systemic vasodilators. Isoflurane dilates pial and dural vessels in a dose-dependent manner[46], thereby increasing perfusion pressure and altering intracranial hemodynamics[47]. Taken together, the rise in RBC velocity under isoflurane anesthesia likely reflects a combination of glucose-dependent flow regulation and site-specific vascular reactivity.

The hormone-secreting capacity of human islet grafts placed on the dura mater, evidenced by glucose-dependent increases in plasma human C-peptide, confirmed functional integration into host glucose metabolism. This validates that the recorded pancreatic β-cell $[Ca^{2+}]_i$ dynamics are physiologically relevant and reflective of insulin secretion. The cranial window enabled stable, long-term imaging, permitting single-cell tracking of $[Ca^{2+}]_i$ activity in vivo. Our analysis focused on established indicators of β-cell function: the period and amplitude of slow oscillations[48], intercellular synchronization[49], and plateau fraction, the proportion of time cells maintain elevated $[Ca^{2+}]_i$ during oscillatory cycles[50]. Parameters widely recognized as hallmarks of coordinated islet activity and glucose responsiveness. 1 h recordings of β-cell $[Ca^{2+}]_i$ dynamics in transiently anesthetized animals revealed pronounced differences between awake and isoflurane-exposed states. In awake mice, dura mater-engrafted β-cells exhibited robust slow $[Ca^{2+}]_i$ oscillations with high intercellular synchronization, indicative of coordinated and functional islet activity. In contrast, isoflurane attenuated $[Ca^{2+}]_i$ oscillations, altered oscillatory patterns, and reduced β-cell network synchronization. The reduction of β-cell $[Ca^{2+}]_i$ signaling under isoflurane anesthesia is likely mediated by inhibition of voltage-gated $Ca^{2+}$ channels[51], interference with ryanodine receptor 1 leading to reduced endoplasmic reticulum $Ca^{2+}$ release[52], or impaired mitochondrial ATP production[53]. Although these adverse effects have been documented in other cell types, their precise contribution to β-cell $[Ca^{2+}]_i$ suppression remains to be fully elucidated. In β-cells in particular, isoflurane is known to enhance $K_{ATP}$ channel activity[54], thereby impeding glucose-induced depolarization and subsequent $Ca^{2+}$ influx. In our experiments, isoflurane anesthesia consistently attenuated β-cell $[Ca^{2+}]_i$ oscillations, reduced the plateau fraction, and impaired insulin secretion, consistent with anesthesia-induced hyperglycemia reported in rodents[55] and humans[56]. These findings indicate that isoflurane disrupts the coupling between $[Ca^{2+}]_i$ dynamics and insulin release, likely by interfering with β-cell $Ca^{2+}$ signaling pathways. This disruption appears to act first on glucose sensing, subsequently dampening glucose-induced $Ca^{2+}$ oscillations. Isoflurane interference with $K_{ATP}$ channel function likely lowers the β-cell glycemic set point and reduces glucose-induced excitability, blunting insulin secretion under increased metabolic demand. Consequently, constant blood glucose under isoflurane fails to maintain β-cell activation, leading to downregulation of glucose-induced $Ca^{2+}$ oscillations.

In contrast, sedation with Hypnorm preserved glucose-stimulated insulin secretion and maintained $[Ca^{2+}]_i$ activity at levels comparable to the awake state, indicating that this anesthetic protocol is less disruptive for in vivo β-cell imaging. Nonetheless, Hypnorm's central suppression of autonomic outflow[57] may still influence islet function, given the critical role of autonomic input in coordinating β-cell activity[58]. Together, these findings underscore the importance of selecting anesthesia protocols that maintain physiological β-cell function during intravital imaging.

## Islet $Ca^{2+}$ dynamics in awake mice under elevated metabolic demands

To examine β-cell $[Ca^{2+}]_i$ dynamics under increasing metabolic demand, we employed subcutaneous glucose injection in awake animals. This approach induced a slow, sustained rise in systemic glucose, allowing assessment of islet function under physiologically relevant conditions. By correlating site-specific glucose distribution and host metabolic responses with β-cell $[Ca^{2+}]_i$ activity, we captured the relationship between progressively increasing secretory demand and islet graft function. Over 90 min of imaging, including a 60 min period of glucose stimulation, β-cells exhibited robust, highly synchronized $[Ca^{2+}]_i$ oscillations with periods of 2–3 min, indicative of strong intercellular coupling and coordinated network activity[59]. This pattern mirrors previously described pulsatile electrical activity and insulin secretion, underscoring the role of rhythmic $[Ca^{2+}]_i$ dynamics in endocrine function[59]. Pulsatile insulin release has been demonstrated in vivo[60] and shown to be synchronized at the single-cell level using fluorescent zinc indicator in externalized pancreas studies[61].

As glucose gradually increased, we observed a progressive reduction in oscillation amplitude and period, accompanied by a modest but significant rise in plateau fraction, while intercellular synchrony remained high. Control experiments ruled out photobleaching or phototoxicity, confirming that these changes reflect a genuine functional adaptation. This transition likely reflects a shift in regulatory dynamics, consistent with dual or integrated oscillator models in which glycolytic feedback modulates slow oscillations via ATP/ADP fluctuations, while membrane excitability governs faster $Ca^{2+}$ rhythms[62]. At elevated glucose concentrations, increased metabolic drive may prolong membrane depolarization, compressing the active and silent phases of the oscillatory cycle and thereby reducing both amplitude and period of the $Ca^{2+}$ signal. Whether this adaptation aligns with predictions from integrated oscillator or Chay–Keizer models[62] remains unresolved, particularly given limitations in temporal resolution that preclude detection of faster $Ca^{2+}$ oscillations. The absence of a first-phase $Ca^{2+}$ peak, commonly observed after intravenous glucose administration[10], likely reflects the gradual kinetics of subcutaneous glucose delivery, which does not acutely trigger glycolytic or membrane excitability thresholds. Our imaging approach was also biased toward slow oscillations, which limited detection of rapid $Ca^{2+}$ transients associated with first-phase insulin secretion[63]. Nonetheless, the persistence of 2–3 min $Ca^{2+}$ oscillations with high synchrony supports the metronome hypothesis[64], proposing that β-cell networks maintain a stable temporal rhythm to coordinate pulsatile insulin release. In this framework, rising glucose prolongs the active phase of the $Ca^{2+}$ cycle, increasing insulin pulse amplitude without altering frequency[64]. The increase in plateau fraction during glucose stimulation reflects prolonged β-cell $[Ca^{2+}]_i$ elevations, promoting $Ca^{2+}$-dependent exocytosis of insulin vesicles to meet heightened metabolic demand. Future studies with higher temporal resolution could detect fast $Ca^{2+}$ oscillations[63], resolve electrically coupled β-cell dynamics, and clarify the roles of leader[9] and hub[8] cells. Three-dimensional imaging may reveal functional heterogeneity across islet layers, while pharmacological interventions (e.g., diazoxide, tolbutamide, GLP-1 analogs) in awake mice could disentangle contributions of metabolic and electrical oscillators.

## Perspective

Using the Mobile HomeCage platform, we established an anesthesia-free intravital imaging approach that enables stable, long-term, and physiologically relevant monitoring of islet grafts. We identified the dura mater as a suitable transplantation site for human and mouse pancreatic islets, offering exceptional optical access and mechanical stability. This approach paves the way for high- and super-resolution microscopy in awake animals (e.g., SIM, STORM, STED) and the potential to visualize subcellular processes using targeted biosensors.

Our study highlights the physiological alterations caused by anesthesia and provides a reliable alternative, enabling physiologically accurate in vivo imaging of micro-organs in awake mice with high stability.

## Methods

### Mice

All animal experiments were approved by the Swedish Animal Welfare Council (Ethical permit: 05480-2023) and conducted in accordance with the "Guide for the Care and Use of Laboratory Animals", eighth edition (2011).

Isolated pancreatic islets were obtained from adult double-heterozygous Ins1Cre:GCaMP3 mice, in which the $Ca^{2+}$ indicator GCaMP3 is specifically expressed in insulin-producing β-cells. For this breeding, the mouse lines B6.Cg-*Ins1*$^{tm1.1(cre)Thor}$/J and B6.Cg-*Gt(ROSA)26Sor*$^{tm38(CAG-GCaMP3)Hze}$/J were acquired from Jackson Laboratories. The genetic background was maintained on a C57BL/6J background by continuous backcrossing for more than 10 generations. Female C57BL/6J mice served as recipients for syngeneic transplantation of mouse islets, while female Rag1$^{-/-}$ (B6.129S7-*Rag1*$^{tm1Mom}$/J) mice were used for xenogeneic transplantation of human islets. Only female mice were used as recipients to facilitate group housing; sex was not considered as a biological variable in this study. Mice were housed in a specific pathogen–free (SPF) facility under controlled environmental conditions, including a 12 h light/dark cycle (lights on at 07:00 and off at 19:00), an ambient temperature of 21–23 °C, and relative humidity of 45–65%. Animals had *ad libitum* access to standard chow diet (2918 Teklad Irradiated Global 18% Protein Rodent Diet) and water.

### Pancreatic islet isolation

Pancreatic islets for $[Ca^{2+}]_i$ imaging were isolated from adult double-heterozygous Ins1Cre:GCaMP3 mice. Pancreatic islets for RBC velocity imaging were isolated from adult non-transgenic littermates derived from the Ins1Cre:GCaMP3 breeding. Donor animals were euthanized by cervical dislocation in accordance with approved ethical protocols. Pancreatic islets were isolated by collagenase digestion of the pancreas, followed by manual selection under a stereomicroscope. A total of 2.5 mL of ice-cold digestion solution containing collagenase A (1.5 mg/mL) in Hank's Balanced Salt Solution (HBSS) supplemented with 0.2% (w/v) BSA in HEPES (25 mM, pH 7.4) was injected intraductally through the cannulated common bile duct. The infused pancreas was transferred to ice-cold HBSS and digested at 37 °C for ~25 min, with intermittent gentle manual agitation until the pancreatic tissue was macroscopically dissociated. The resulting suspension was further disaggregated mechanically using a Pasteur pipette. Isolated islets were manually selected, washed twice with cold HBSS, and cultured in RPMI-1640 medium supplemented with 10% (v/v) fetal bovine serum, 2 mM L-glutamine, 100 U/mL penicillin, and 100 µg/mL streptomycin (all from Gibco). Islets were maintained in a humidified incubator at 37 °C with 5% $CO_2$), and the medium was changed twice weekly for at least 1 week prior to transplantation.

### Cranial surgery and islet transplantation onto the dura mater

Cranial surgery and islet transplantation were performed under sterile conditions, using autoclaved instruments and materials on an air-exhaustion bench. Female C57BL/6J or Rag1$^{-/-}$ mice ( > 12 weeks of age, >20 g body weight) were used. Anaesthesia was induced with 3% isoflurane (Baxter Medical AB) in oxygen using an isoflurane vaporizer (Anaesthesia Unit 400; Univentor) and maintained at 1–1.5% isoflurane. Mice were positioned in a stereotactic frame (Narishige; Japan) equipped with a breathing mask, heated platform (37 °C), and head fixation using two ear bars. Buprenorphine (Temgesic; Indivior Europe) was administered subcutaneously (s.c.) at 0.1 mg/kg body weight for analgesia. Ocular surfaces were protected with ophthalmic gel (Viscotears; Thea). The scalp was shaved using a trimmer and razor blade. The shaved skin was topically treated with lidocaine

hydrochloride (10%, Xylocaine; Aspen) for local anesthesia. After a circular midline scalp incision (1–1.5 cm diameter), the skin over the parietals was removed to expose the cranium. Cranial bones and injured skin were treated several times with iodine solution (15 mg/mL; Nyodex) and dried with cotton swabs. The periosteum was removed with forceps and microscissors until the bone was completely bare. A thin layer of cyanoacrylate tissue adhesive (Vetbond; 3 M) was applied to seal the incision surrounding the exposed skull. A circular craniotomy with a maximum diameter of 5 mm was performed over the parietal bone, adjacent to the central sagittal suture. Drilling was performed using a micromotor hand drill (HP4-917, MH-170; Foredom) equipped with a steel bur head (Densply 006; Maillefer Instruments). The drilling site was irrigated with compressed air to remove debris and prevent bone overheating. Finally, the detached bone fragment was carefully lifted out with hooked forceps, exposing the dura mater. A small dural puncture was created using two 27 G needles to permit transient cerebrospinal fluid release into the cranial window during the postoperative period. The exposed meninges were immediately irrigated several times with cold (4 °C) sterile Ringer's solution (in mM: 125 NaCl, 2.5 KCl, 1.25 $CaCl_2$, 25 $NaHCO_3$, 10 glucose; pH 7.4) and remained covered until transplantation with islets. Ten pancreatic islets were transferred from RPMI medium into pre-warmed (37 °C) Ringer's solution and carefully positioned onto the dura mater in a ~5 µL droplet using a 20 µL pipette. A glass coverslip (5-6 mm diameter, Menzel #1; ThermoFisher Scientific) was gently placed over the cranial window using fine forceps. While holding the coverslip in place with a cotton-tipped applicator, instant adhesive (Loctite 401; Henkel) was applied around its edges to bond it to the skull. The adhesive was allowed to cure for 5 min under gentle pressure. Exposed bone surrounding the cranial window was coated with a 2:1 mixture of dental cement (cold-curing acrylic denture repair material; Vertex) and instant adhesive (Loctite 401; Henkel). A four-armed metal head plate (model 3 or 4; Neurotar) was immediately embedded over the coverslip. The dental cement was allowed to cure for at least 1 h, depending on the setting time. Before terminating anesthesia, 200 µL sterile saline (0.9% NaCl; Fresenius Kabi) was administered s.c. For the following 3 days, buprenorphine (Temgesic; Indivior Europe) was administered s.c. at 0.1 mg/kg body weight, twice daily. The skin surrounding the cranial window was treated with Durahesive Skin Barrier (Silesse; ConvaTec) during this period.

### Human pancreatic islet transplantation

For xenotransplantation experiments, immunodeficient Rag1$^{-/-}$ mice from an in-house breeding colony at Karolinska Institutet were used as recipients. Human pancreatic islets were procured through the Nordic Network for Islet Transplantation from deceased organ donors. The protocol for human pancreatic islet procurement and use was approved by the Regional Ethics Review Board in Stockholm (Regionala etikprövningsnämnden i Stockholm; approval number 2006/515-31/3). Informed consent was obtained from donors' next of kin, and donor information was anonymized to ensure ethical compliance and privacy protection. Human islets were isolated by enzymatic digestion with collagenase, followed by density gradient purification. Upon receipt, islets were maintained in CMRL-1066 medium (Gibco) supplemented with 5.6 mM glucose (Sigma-Aldrich), 10% (v/v) fetal bovine serum, 2 mM L-glutamine, 100 U/mL penicillin, 100 µg/mL streptomycin, 10 mM HEPES, 10 mM nicotinamide, and 5 mM sodium pyruvate (all from Gibco) in a humidified incubator at 37 °C with 5% $CO_2$. Human islet quality was assessed 1 and 3 weeks after arrival by measuring glucose-stimulated $Ca^{2+}$ activity and insulin secretion (Supplementary Fig. 1B–C). Four weeks after arrival, each animal received a transplantation of 50 human pancreatic islets onto the dura mater. Cranial window surgery and transplantation followed the protocol established for syngeneic mouse-to-mouse transplantation, with one modification: to stabilize individual islets during transfer and sealing, a

sterile 2.5% gelatin (porcine skin; Sigma-Aldrich) solution in Ringer's was used during placement on the dura mater. This approach ensured graft retention and positional stability during and after coverslip application. Donor characteristics: female; age range 70–75 years; body mass index (BMI) 20–24 kg/m²; HbA1c 30–40 mmol/mol.

### Intravital microscopy in awake and anesthetized animals

**Awake imaging.** Mice were habituated to the Mobile HomeCage system (Neurotar, Helsinki) starting 3 weeks after transplantation, undergoing 5 daily 1 h sessions with the cranial window secured in the head holder. For imaging, the cage was positioned under the microscope, with the cranial window stably aligned to the objective. Mice were free to move their body and limbs, with the air-cushioned cage compensating for motion. Imaging was performed under low-light, low-noise conditions. Confocal microscopy was conducted on a Leica TCS SP5 II system with a 25× water-immersion objective (NA 0.95; Leica Microsystems). $[Ca^{2+}]_i$ time-series recordings were acquired from a single confocal plane at 1 frame per second (fps), under basal conditions or after s.c. glucose administration (2 g/kg body weight). Glucose distribution after s.c. injection was assessed using 2-NBDG (2-(N-(7-nitrobenz-2-oxa-1,3-diazol-4-yl)amino)−2-deoxyglucose; ThermoFisher Scientific) at 0.25 mg/animal. Blood vessels were visualized by intravenous (i.v.) injection of either TMR-dextran (tetramethylrhodamine-dextran, 2000 kDa; ThermoFisher Scientific) at 0.5 mg/animal or lectin-649 (tomato lectin LEL, DyLight 649; ThermoFisher Scientific) at 0.1 mg/animal. Blood glucose levels were measured from tail vein samples using a handheld glucometer (Accu-Chek Aviva; Roche Diagnostics), before and after cage placement.

**Anesthetized imaging.** Imaging during the first 4 weeks after transplantation was performed exclusively under isoflurane anesthesia, whereas subsequent $[Ca^{2+}]_i$ imaging (6-16 weeks post-transplantation) was conducted under either isoflurane or Hypnorm anesthesia. Isoflurane anesthesia was induced at 3% isoflurane (Baxter Medical AB) and maintained at 1–1.5% with oxygen using a vaporizer system (Anaesthesia Unit 400; Univentor). For the 1 h transient anesthesia recordings, 2% isoflurane was continuously administered. Hypnorm anesthesia (fentanyl–fluanisone 0.6/20 mg/kg; VetaPharma, plus midazolam 10 mg/kg; Panpharma) was administered i.p. with supplemental oxygen. In both protocols, animals were maintained on a temperature-controlled heating pad (37 °C).

### Quantification of revascularization and vascular permeability

Revascularization of dura mater-engrafted islets was quantified from confocal image hyperstacks acquired with a z-step size of 1.5 μm, using Volocity software (PerkinElmer). The graft volume was defined by the GCaMP3 fluorescence signal, which delineated the β-cell mass. Capillary volume and diameter within the graft were calculated using TMR-Dextran (2000 kDa)-labeled vasculature after i.v. injection, applying uniform thresholding parameters in Volocity. Relative vascularization was expressed as the ratio of vessel volume to total graft volume, assessed at 1, 2, 3, 4, and 10 weeks post-transplantation by longitudinal imaging of the same animals. Three-dimensional reconstruction of i.v. injected lectin-649-labeled vasculature at 5 weeks post-transplantation was performed from confocal hyperstacks acquired with a z-spacing of 1.5 μm and visualized with the Volume Viewer plugin (v2.01, ImageJ; NIH). Vascular permeability was assessed 6 weeks after transplantation by dual-fluorescence intravital imaging following i.v. injection of two fluorescent dextrans of differing molecular weights: TMR-Dextran (tetramethylrhodamine-dextran, 2000 kDa; ThermoFisher Scientific) and CB-dextran (cascade blue-conjugated dextran, 10 kDa; Thermo-Fisher Scientific), each administered at a dose of 0.5 mg/animal. Transcapillary leakage was monitored over a 12.8 min period by sequential imaging at two focal planes (separated by 14 μm) with a temporal resolution of 14 s. Fluorescence intensity in the extravascular

compartments of the dura mater and islet grafts was measured for both channels. After subtracting baseline signals (before injection), vascular permeability ratio was defined as the extravascular fluorescence intensity of the 10 kDa dextran, normalized to the 2000 kDa reference dextran at each time point post-injection.

### Immunohistochemistry

Cortical and meningeal tissues bearing surface-engrafted pancreatic islets were dissected and fixed overnight in 10% neutral-buffered formalin. Samples were cryoprotected by sequential incubation in a graded sucrose series (10–30%) and embedded in Tissue-Tek (Sakura Finetek Europe) prior to sectioning. Cryosections (15 μm thickness) were permeabilized in 0.5% Triton X-100 in PBS and blocked with 10% (v/v) fetal bovine serum in PBS for 1 h at room temperature. Sections were incubated overnight at 4 °C with the following primary antibodies: rat anti-insulin directly conjugated with Alexa Fluor 488 (1:100; R&D Systems), guinea pig anti-VAChT (vesicular acetylcholine transporter, VAChT; 1:500; Synaptic Systems), and rabbit anti-TH (tyrosine hydroxylase, TH; 1:500; Millipore). After PBS washes, appropriate secondary antibodies were applied for 1 h at room temperature: goat anti-guinea pig Alexa Fluor 546 (1:500; ThermoFisher Scientific) and goat anti-rabbit Alexa Fluor 633 (1:500; ThermoFisher Scientific). Sections were mounted using ProLong Gold Antifade Reagent with DAPI (ThermoFisher Scientific) and imaged using a Leica TCS-SP5 II confocal microscope.

### RBC fluorescent labeling and imaging

Red blood cells (RBCs) were labeled with the far-red lipophilic carbocyanine dye DiD solid (1,1'-dioctadecyl-3,3,3',3'-tetramethylindodicarbocyanine,4-chlorobenzene-sulfonate salt; Invitrogen) following a modified protocol[65]. Seventy microliters of blood were withdrawn from the arterial line of a C57BL6/J mouse. RBCs were isolated by adding 140 μL of blood plasma buffer (BPB; 128 mM NaCl, 15 mM glucose, 10 mM HEPES, 4.2 mM $NaHCO_3$, 3 mM KCl, 2 mM $MgCl_2$, and 1 mM $KH_2PO_4$, pH 7.4) and centrifuged for 5 min at 240 × g. Red cells were resuspended in 280 μL BPB and mixed gently in a dye solution of 280 μL Diluent C (CGLDIL; Sigma-Aldrich) and 7 μL DiD solid (2.5 mg/mL in 100% ethanol). The mixture was incubated at 37 °C for 10 min with periodic inversion to ensure uniform labeling. After labeling, 140 μL of mouse serum (50 mg/mL; Invitrogen) was added, and the suspension was incubated for an additional 1 min. Fluorescent RBCs were centrifuged at 240 × g for 5 min and washed twice by careful mixing in 700 μL BPB containing 10% mouse serum, followed by a final centrifugation step (5 min, 240 × g) to remove unbound dye. Labeled erythrocytes were resuspended in 200 μL BPB, and 100 μL was injected intravenously into the tail vein of each recipient animal. To visualize islet vasculature during RBC velocity measurements, mice received an i.v. injection of FITC-dextran (fluorescein-conjugated dextran, 2000 kDa; ThermoFisher Scientific) at a dose of 0.5 mg/animal. RBC velocity was acquired using a Leica TCS-SP5 II microscope equipped with a resonance scanner. Time-series recordings of RBCs were obtained from single confocal planes at a frame rate of 55.5 fps.

### Analysis of β-cell $[Ca^{2+}]_i$ dynamics

Pancreatic β-cell $[Ca^{2+}]_i$ traces were extracted from GCaMP3 fluorescence images using Fiji (ImageJ, latest version). Regions of intracellular $Ca^{2+}$ signals from single β-cells were manually outlined, and changes in GCaMP3 fluorescence intensity were registered over the time series. For each individual cell, basal GCaMP3 fluorescence ($F_0$) was determined by averaging the lowest 25% of intensity values. Fluorescence intensity was normalized to baseline ($F_0$) and expressed as $\Delta F/F_0$. Oscillations of normalized GCaMP3 traces were analyzed by power spectral analysis using a custom MATLAB (MathWorks) script[66] adapted to differentiate slow oscillations, defined as oscillations lasting 60–600 s. Dominant periods of slow oscillations were determined from fast Fourier transform (FFT) power spectra. Amplitude values

were calculated as the square root of total power for slow oscillations. Correlation analyses between the normalized GCaMP3 traces of all selected β-cells within an imaged islet were performed using Pearson's correlation in MATLAB. The total islet correlation strength for each condition was calculated as the mean of all positive pairwise correlation coefficients (*r*), excluding autocorrelations. Cartesian coordinates of selected β-cells were used to construct topographic representations of β-cell pair correlations in MATLAB. Paired β-cells were connected by color-coded lines corresponding to correlation ranges: blue ($r = 0.10-0.25$), green ($r = 0.25-0.50$), yellow ($r = 0.50-0.75$), and red ($r = 0.75-1.00$). The plateau fraction was used as a metric for quantifying the proportion of time cells remained in an active state. For each oscillation, the half-maximal amplitude was defined as the activation/deactivation threshold. Onset and termination points of oscillations were manually annotated in MATLAB. The plateau fraction was calculated as the cumulative duration of all oscillatory events divided by the total recording time.

### Glucose tolerance tests

Mice were fasted for 6 h during the light phase prior to the glucose tolerance test. Blood glucose was measured at baseline (0 min) and at 15, 30, 60, 90, and 120 min following intraperitoneal (i.p.) or subcutaneous (s.c.) glucose administration (2 g/kg body weight). Glucose concentrations were determined with an Accu-Chek Aviva glucometer (Roche Diagnostics, Scandinavia AB). During the test, blood samples were collected into Microvette CB 300 $K_2$ EDTA tubes (SARSTEDT AG & Co. KG) and kept on ice. Samples were centrifuged at $5000 \times g$ for 30 min at 4 °C, and plasma was harvested and stored at −80 °C until biochemical analysis.

### Biochemical blood plasma assays

Mouse insulin concentrations were quantified using an ultrasensitive mouse insulin enzyme-linked immunosorbent assay (ELISA) (Crystal Chem, Inc.). Mouse C-reactive protein (CRP) was measured with a mouse-specific ELISA (Crystal Chem, Inc.). Mouse tumor necrosis factor-alpha (TNF-α) and interferon-gamma (IFN-γ) were quantified using mouse-specific ProQuantum immunoassays (Invitrogen). Human C-peptide was measured using human-specific ELISA (Crystal Chem, Inc.). All assays were performed following the manufacturer's instructions.

### Statistical analysis

Data were analyzed using GraphPad Prism (GraphPad Software). Normality was assessed with the Kolmogorov-Smirnov test. Statistical comparisons between two paired, normally distributed groups were performed with two-sided paired *t*-tests. Comparisons among three or more groups were evaluated by one-way analysis of variance (ANOVA) followed by Tukey's post-hoc test. Time-course datasets (vascular permeability, transient isoflurane 1 h $[Ca^{2+}]_i$ recordings, and glucose-stimulated $[Ca^{2+}]_i$ responses) were analyzed using two-way ANOVA with Bonferroni post hoc correction. Data are presented as mean ± s.e.m. Differences were considered statistically significant at $P < 0.05$. No statistical method was used to predetermine sample size. No data were excluded from the analyses. The experiments were not randomized. The Investigators were not blinded to allocation during experiments and outcome assessment.

### Reporting summary

Further information on research design is available in the Nature Portfolio Reporting Summary linked to this article.

## Data availability

The data supporting the findings of this study are available within the paper and its Supplementary Information files. Source data are provided with this paper.

## Code availability

Custom code used for data analysis in this study is available from the corresponding author upon request and is described in the Reporting Summary.

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

## Acknowledgements

P.-O.B. acknowledges funding support from the European Research Council (ERC) "Advanced Grant" (2018-AdG 834 860 EYELETS), the Family Erling-Persson Foundation, the Jonas & Christina af Jochnick Foundation, the KI Foundation, the Novo Nordisk Foundation, the Swedish Diabetes Association, the Swedish Foundation for Strategic Research (SSF MED-X 2018), and the Swedish Research Council. The authors thank Neurotar (Helsinki, Finland) for their support and for providing a demonstration of the Mobile HomeCage system. Human pancreatic islets were obtained from the Nordic Network for Clinical Islet Transplantation, supported by the Swedish national strategic research initiative Excellence of Diabetes Research in Sweden (EXODIAB).

## Author contributions

P.T. designed the study, performed surgical procedures and islet transplantations, conducted intravital microscopy and immunohistochemistry experiments, analyzed data, and prepared the original draft and figures. M.V. performed intravital microscopy and metabolic experiments, analyzed $[Ca^{2+}]_i$ imaging data, and contributed to manuscript editing. I.V.-A. performed intravital microscopy and metabolic experiments, analyzed data, and contributed to manuscript editing. M.K. provided technical guidance and contributed to manuscript revisions. P.-O.B. conceived the study, supervised the project, and revised the manuscript.

## Funding

## Competing interests

P.-O.B. is co-founder and CEO of Biocrine, a biotech company that is focusing on Apolipoprotein C-III as a potential drug target in diabetes. The rest of the authors declare no competing interests.
