## [Transparent Peer Review file · Nature Communications]

Stable intracranial imaging of dura mater-engrafted pancreatic islet cells in awake mice

Corresponding Author: Dr Philip Tröster

Version 0:

Reviewer comments:

Reviewer #1

(Remarks to the Author)

This team describe an important new method for imaging transplanted tissue in awake, unanaesthetised mice. Aside from the immense technical feat, this is worthy of note because it overcomes one of the major limitations of other longitudinal imaging approaches that require anaesthesia, which very obviously affects the behaviour of tissues, particularly excitable ones. Furthermore, this study is likely to be adopted and highly cited since, it will apply not only to the study of islets but feasibly a range of other tissues/organoids.

I have the following comments:

In the introduction the authors imply that all isoflurane experiments are flawed when interpreting calcium data. I'm not convinced this is true as experiments done at low anaesthetic depths and before the animals become hyperglycaemia are arguably still admissible.

Figure 1G is really remarkable and it would be worth mentioning in the body of the text that this time composite image means that 10 minutes worth of recordings at 1 frame per second are superimposed here with no visible blur. Is this true for all images and is there a mathematical way of representing this eg by averaging displacement from the central (time course) image?

Supplemental Figure 1 – again this is a reassuring dataset since stress-induced hyperglycaemia is an obvious concern here. Please add in a Supp 1b to show the average (and range) blood sugar levels for each type of imaging session (eg awake vs immobilised) in absolute terms (mM or mg/dl).

Line 205: "Vascularization and innervation of pancreatic islets at the brain surface"

"The brain-transplanted pancreatic islets integrated well and rapidly on the brain surface and showed excellent survival when placed centrally not too close to the border of the cranial window."

Either here or in methods please clarify – were you attempting to implant only one islet per brain window? What was the implantation success rate?

Figure 2A-G - % islet volume covered with blood vessels seems an unusual way of describing the implantation process. There is insufficient data here to convince the reader that the blood vessels are extending throughout (growing into not simply overlaying) the islet and that their calibre/density is redolent of the native pancreas. In the Conclusion some discussion should be made about the blood supply of the dura (presumably the meningeal arteries, the venous drainage as well as the origin of the nerve supply)

The data implying innervation is basic in the sense that there is no attempt to compare the density or distribution of the innervation pattern with respect to this group's well established eye model or the endogenous pancreas. For example the TH and VAChT images look to be peripheral and sparse. Given that some of the dural innervation comes from the vagus, this is a missed opportunity to show that this model may well have strengths in terms of the origin of the innervation. This would be an important study in the future, in order to cement this platform as a good model for studying islets.

Blood flow videos: supp video 1 and 2 are BOTH from awake animals? Why did you not provide a comparator from anaesthetised experiment?

Were these velocity experiments performed on a single animal once awake and once under anaesthesia?

Without this info it's difficult for me to comment more at this stage however:

1. These images are extremely impressive and I commend the experimentalists for their work
2. Isoflurane is well documented to increase cortical blood flow velocity so this at first glance makes sense. However, the animals were obviously deeply anaesthetised for an hour. This means they were also likely hypercapnoeic (a confounder) and this doesn't represent the depth or duration of anaesthesia for other islet imaging experiments eg in the eye.
3. If blood vessel calibre is not changed, why is RBC flow faster? I see that you have tried to conclude on this in the Discussion but I think that the conclusions made there are too bold and no consideration of the myriad of other confounders eg depth of anaesthesia have been considered.

Calcium imaging section

Supplemental video 3 is a fantastic achievement.

Visual assessment of this video looks different to calcium waves that have previously been imaged in vitro and in vivo. There is no obvious starting zone/leader area or "wavefront" and I wonder if this is because of the speed of the GCaMP3 and your frame rate – please comment (work of Paukert springs to mind comparing GCaMP 3 and 6 in being able to distinguish neuronal wave patterns).

I agree that amplitude decreases with anaesthesia. I'm not sure I agree with your conclusions about periodicity (which is not defined in your methods). What is happening is that a few of the beta cells that were already dark (cells at the bottom of your image 4A) fail to reach your threshold for activation and therefore become "disconnected" – the Pearson heatmaps show very clearly that the same cells (in an offset cross pattern) that are already poorly connected in the awake setting become darker blue when anaesthetised because of dampened beta cell activity globally. I therefore find the connectivity analyses here unhelpful in describing what you are observing. Fundamentally the pattern of connectivity is preserved it's just that the entire signal is dampened.

For the experiment comparing different anaesthetic agents – you mention the "same islet grafts" were repeatedly imaged – please clarify how many different islets/animals? The cross section that you choose to display does not look identical in the three different conditions (isoflurane one looks different).

Please define how you analysed wave parameters such as amplitude and "period" in methods. It also looks like your duty cycle (probably a better measure of insulin secretion and roughly the product between frequency and duration) is altered in anaesthesia. For a broader audience it would also be helpful to define what you mean by slow and rapid oscillations in your methods. For example in Fig 4b there clearly are fast oscillations superimposed on the slower waves in the isoflurane condition but I'm not sure if you are reporting them as I do not know what your cut off is for defining a "fast wave"

Line 322 "Compared to non-anaesthetised mice, the degree of synchronization deviate more under isoflurane anaesthesia than under administration with Hypnorm" I disagree with this. Firstly your "exemplar" comparisons clearly shows that hypnorm results in a much poorer average R value – much more blue! The combined data in 4J is not conclusive and when analysed using an ANOVA with post hoc correction the blue and orange bars are NOT different.

The attempt to define leaders is interesting but the data too weak to draw conclusions from. There is no attempt to quantify "shift" in terms of their identity or position related to baseline/non anaesthetised. Also, without comparator data from the same length of imaging in animals awake the whole time it is impossible to tell whether this shift wouldn't have happened anyway. In fact, as supp Fig 4 shows, the zonal location of your first responders is actually very well preserved throughout anaesthesia.

Granger – given that you have already concluded that isoflurane reduces the number of connected cells it's hardly surprising that the Granger associations are also reduced. I don't think this adds much I'm afraid.

DISCUSSION

This is the first mention of meningeal puncture to allow CSF leak to bathe the islets – if this is central to the success/survival of the transplants then it needs to be very clearly described in the surgical Methods (presumably a single puncture hole with a ?G needle)

This model is being posited as a platform for tissue modelling without anaesthesia. This is very exciting but there is no discussion as to whether other transplanted tissue types have been attempted or if there is evidence that they are likely to engraft?

Were you able to detect calcium oscillations in every islet in every imaging attempt? How often were you able to image calcium oscillations in all elements of the imaging protocol (pre, induction, anaesthetised, recovery?) The data is presented in such a way that this is always measurable – is that true?

I think it might be worth noting that overall patterns of connectivity do seem to be preserved with isoflurane imaging and that the comparison of different islet stimuli that are imaged and compared under isoflurane will still have relevance, latest caveats accepted.

Reviewer #2

(Remarks to the Author)

The aim of this study is to set up a new technique to image pancreatic islet cells in awake mice. To overcome the fact that the pancreas is anatomically located deep in the abdominal cavity, which presents as a significant difficulty on intravital microscopy, the authors transplanted the islet cells onto the surface of the brain to improve imaging accessibility. Additionally, the authors described the measurement of calcium dynamics in these transplanted cells, as well as blood flow

velocity of the islet organoids, in awake mice with head fixation. There are many critical concerns with the methodology and experimental design that need to be addressed prior to publication:

Major:

- 1) The surgical removal of the skull to generate a transparent cranial window will create the bulging of the brain through the opening as a response from existing intracranial pressure. Arguably, this is very far from "non-invasive" as the title suggested. As such, this term should be removed entirely from the manuscript. What are the implications on brain integrity, stability and inflammation?
- 2) The transplanted organoid is placed onto the dura. How does this impact on the perfusion of the dura? Moreover, what is the effect of the cerebral spinal fluid on the 'micro-organs', or in this case, the transplanted islets versus a naïve pancreas?
- 3) On the same train of thought, the authors demonstrate positive staining for enzymes for parasympathetic and sympathetic neurons in transplanted cells. How is the ratio/composition compared to the pancreatic ones in situ? More importantly, are these neurons functional?
- 4) Successful islet perfusion was shown, but how about endothelium activation/integrity markers? Does the transcriptional signature of the islet vasculature match that of the brain, or the pancreas?
- 5) Whilst the brain has an anatomical advantage that allowed the authors to imaging this tissue bed when the animal is awake, the study design ignored the biological implication on this very complicated and rather unique organ. The brain itself does not harbour any pain receptors which is an advantage for transplantation studies, however the removal of the skull and the associated surgical manipulation will induce pain. Was any analgesia used beyond day 3 post-surgery? The administration of analgesia will impact on inflammation and subsequently blood flow.
- 6) Furthermore, organoid/cell transplantation in immune-competent animals will generate a host response that is highly inflammatory which impact on blood flow. These factors need to be considered and measured prior, during and after to transplantation/imaging.
- 7) The authors were very keen to showcase their analysis of calcium oscillation, however calcium signalling suggests the cell is alive but not necessarily functional. The insulin production/cell should be shown from the islet vs an islet in the pancreas in situ.
- 8) Similarly, the study design does not allow for biological-relevance comparison. What happens to the islet perfusion and their function when mice are eating, or recently had a meal? What happens when the mice have an administration of sugary drinks?
- 9) The authors are making us aware that isoflurane will inevitably cause hyperglycaemia, but how biologically relevant is this (especially if imaging is not performed chronically)? In other words, assuming the imaging is done acutely on multiple timepoints within the same animal, does the blood glucose fluctuate to a level that is considered pathogenic?

Minor:

- 1) Fig 2 A-E is not referred to in text.
- 2) Figure 2 A is a single Z-stack, which seemingly shows the dura vasculature, and does not confirm vascularisation of the islet. Multiple z-stacks should be shown here and 3-D reconstituted to shown integration of vasculature into the islet.
- 3) Fig 2E – what is 100% vascularisation? What is the score 'relative' to?
- 4) Fig 3 – heart rate, blood pressure and full blood counts should be shown here.
- 5) Discussion is not written at an adequate quality, it does not discuss how the study integrate with existing literature.
- 6) Line 407 - which study was being referred to? Their current study did not conduct a metabolic transplantation,... etc.
- 7) Line 420 - in comparison to the neuronal stains, at least the authors did compare how the blood flow velocity in their transplanted islets to the native ones. However, are the transplanted ones now perfusing too fast, assuming the in situ measurements quoted (500 um/sec) were done with anaesthetics (ketamine). Their iso-treated blood flow was closer to 1000 um/sec.
- 8) Line 532 – a major overstatement saying all anaesthetic should be avoided during intravital microscopy. This needs to be rephrase and tone down to suggest appropriate controls are needed, and not completely remove anaesthesia.

Reviewer #3

(Remarks to the Author)

Reviewer #4

(Remarks to the Author)

In their paper, the Berggren lab describes a technique for intravitaly imaging brain-transplanted micro-organs in awake animals. There is a critical need in the field for intravital imaging of biological processes in awake animals, as anesthesia can potentially alter physiological states and research outcomes. The researchers have developed a novel microscopy platform that involves transplanting micro-organs, specifically pancreatic islets, onto the brains of mice, allowing for imaging of the tissues in awake animals.

While the method introduced is interesting, the paper could benefit from more explanations and demonstrations of applications. Additionally, some claims require further support. We have the following concerns/questions:

Major Comments:

1. The paper explains well that imaging in awake animals is a significant step forward since current in vivo imaging requires animals to be anesthetized, which can alter tissue physiology and impact experimental outcomes. However, the paper does not adequately describe the disadvantages of their methods, specifically implanting organs in different niches. The brain is a unique organ with a very specific environment, which highly limits the experimental setting. Some biological conclusions that can only be drawn from physiological conditions (i.e., inside the actual organ within a specific niche) would still require the use of different imaging techniques, albeit with different types of anesthesia. In general, the paper focuses solely on the advantages of the newly developed technique and neglects to mention its limitations. This needs to be further discussed in the introduction and/or discussion section of the manuscript. Furthermore, the paper would benefit from explicitly mentioning the specific applications of the method.

2. In line 127, the authors suggest that their technique can be used to image tissues/micro-organs/organoids implanted in the brain. Are these transplantation assays feasible? Can the authors cite other studies that have performed successful transplantations of other tissues in the brain recapitulating the physiological/pathological conditions?

3. The paper claims broad applicability of the technique and differentiates it from existing methodologies. However, such a claim can only be made if they demonstrate multiple applications. For example, manipulating calcium dynamics with well-characterized drugs, such as an acetylcholinomimetic, could illustrate the versatility of the platform for functional assays and enhance the robustness of the authors' claims. Alternatively, transplanting and imaging other tissues would support these claims. If this is not possible in a reasonable timeframe, it would be fair to tone down the claims of broad applicability.

4. In Supplementary Figures 3 and 4, the authors characterize the first responder β -cells under anesthesia in comparison to the awake state. Do the data plotted in these figures originate from multiple mice? If not, can the authors comment on how reproducible these observations are across different mice and whether leader cell properties depend on the localization of the cell within the transplant?

5.
Minor Comments:
1. The authors should consider rewriting the introduction section, as many aspects mentioned there are repetitive and do not add to the manuscript.

2. The authors mention that the time used for imaging the mice in their experimental setup is one hour. Can the imaging session be prolonged based on the scientific question, or is it restricted to short-term imaging? Can the same mouse be imaged multiple times (days, months)?

3. The authors should mention in the methods section how acclimatization is performed in more detail. Were the mice kept in the imaging holder for the same amount of time as the real imaging session on a daily basis? Were stress levels measured after the first imaging session when the mouse was awake before it underwent anesthesia?

4. The authors have shown that the pancreatic islet transplant is vascularized and innervated. Can they show, with whole mount staining, how this compares to the pancreas (organ) under normal homeostatic conditions? This kind of characterization is essential to determine the type of experiments that can be performed using this technique.

5. In Figures 4g-j, the authors describe how oscillation periods and amplitudes differ between awake and anesthesia treatment. Can they highlight the profile of leader cells in these graphs?

Reviewer #5

(Remarks to the Author)

I co-reviewed this manuscript with one of the reviewers who provided the listed reports. This is part of the Nature Communications initiative to facilitate training in peer review and to provide appropriate recognition for Early Career Researchers who co-review manuscripts

Version 1:

Reviewer comments:

Reviewer #1

(Remarks to the Author)

Line 196 – please state in the main text here when you did the permeability experiment after transplantation (we find that nascent blood vessels shortly after implantation are leaky but that improves on full maturation). This is not a criticism it's just useful for readers interested in implantation biology to have context.

Line 246-8 "In the fully anesthetized state, $[Ca^{2+}]_i$ activity was strongly suppressed, characterized by more irregular and attenuated oscillations." It is very clear that the oscillations are altered in anaesthesia but they are fundamentally still measurable and (we know from others' work) still subject to measurable insulinotropic (or at least glucose responsive) modulation. Therefore I am not comfortable with the term "strongly suppressed" because it implies that there is loss of discernible or useful oscillatory data which really isn't true. I would suggest something more like: "In the fully anesthetized state, $[Ca^{2+}]_i$ activity was significantly altered, characterized by more irregular and attenuated oscillations."

The discussion around slow oscillations and the potential for isofluorane to have partially uncoupled calcium dynamics from insulin secretion, with reference to Satin and others, is much improved. I don't think that basal insulin secretion is completely inhibited under isofluorane as the glucose only rises to 12 mM and insulin levels are still detectable. I do agree that this has uncovered evidence that there are probably multiple/redundant metabolic regulators of the slow oscillations (and some are clearly retained under isofluorane).

Once again congratulations on this excellent work.

Reviewer #2

(Remarks to the Author)

Whilst the authors have addressed most of our concerns, there are still some issues we believe refrain the manuscript from being published at this current stage.

1) "Non-invasive" is still mentioned at least thrice throughout the text.

2) Systemic inflammation markers are not indicative of brain inflammation. Brain tissue homogenate should be analysed for inflammatory markers (e.g TNF α , IL-6, IL1b, IFN γ), especially to reflect their claim that the craniotomy conducted does not impact on the brain. Furthermore, brain-resident cell composition and their cytokine production profile should be shown in the regions surrounding the islets vs contralateral control (microglia, astrocytes, neuron).

3) If direct functional proof of neuronal connectivity is beyond the scope of the present study, please do not overstate there is evidence of "innervation of islets".

4) Our concern with transplantation of islets to the brain is not on strain difference of animals (i.e. allogeneic transplantation), but whether the introduction of islets which would be seen as foreign in the brain/dura will induce an inflammatory response and cause an recruitment of immune cells. This relates to our criticism in Point #2 above.

Reviewer #3

(Remarks to the Author)

Reviewer #4

(Remarks to the Author)

We thank the authors for their thoughtful and comprehensive revisions. The updated manuscript is significantly strengthened by the inclusion of new data on vascular integration, metabolic functionality, and a more balanced discussion of both the advantages and limitations of the proposed technique.

The authors have effectively addressed our main concerns and clarified the physiological relevance and translational potential of their approach. The addition of human islet transplantation data and glucose-stimulated functional responses meaningfully expands the scope and impact of the work.

We are pleased to support the publication of this manuscript. As a final suggestion, we encourage the authors to adopt more precise terminology when describing their method. While the term "non-invasive" has been removed from the title, referring to the technique throughout the manuscript as "less invasive" or "minimally invasive" would more accurately reflect the procedural refinement in comparison to other approaches involving deep tissue access or implantable devices.

This is an elegant and technically impressive study that will be of broad interest to those working in longitudinal imaging, transplantation models, and β -cell physiology. We look forward to its publication.

Reviewer #5

(Remarks to the Author)

REVIEWER COMMENTS

Reviewer #1 (Remarks to the Author):

This team describe an important new method for imaging transplanted tissue in awake, unanaesthetised mice. Aside from the immense technical feat, this is worthy of note because it overcomes one of the major limitations of other longitudinal imaging approaches that require anaesthesia, which very obviously affects the behaviour of tissues, particularly excitable ones. Furthermore, this study is likely to be adopted and highly cited since, it will apply not only to the study of islets but feasibly a range of other tissues/organoids.

We sincerely thank Reviewer #1 for his/her time, effort, and thoughtful evaluation of our manuscript. We particularly appreciate the reviewer's insightful comments regarding the Ca^{2+} imaging data, which played a key role in shaping our revisions. In response to these suggestions, we critically re-assessed our Ca^{2+} imaging analysis strategy. Due to the inherent technical challenges and variability associated with certain complex Ca^{2+} signal parameters, we chose to emphasize more robust and reproducible metrics that still provide meaningful insights into islet graft functionality.

Furthermore, we have substantially expanded the physiological characterization of the islet grafts. In particular, we have included new data evaluating vascularization and metabolic integrity, which we believe significantly strengthen the study's overall conclusions. We trust that these revisions address the reviewer's concerns and align with the expectations outlined in the initial review.

I have the following comments:

1. In the introduction the authors imply that all isofluorane experiments are flawed when interpreting calcium data. I'm not convinced this is true as experiments done at

low anaesthetic depths and before the animals become hyperglycaemia are arguably still admissible.

We agree with the reviewer. Our emphasis on hyperglycemia as the primary issue may have inadvertently created a misleading impression. In reality, hyperglycemia, particularly in the absence of glucose stimulation, is not as critical a factor as we initially suggested. Our measurements show an average increase in blood glucose of approximately 3–4 mmol/L after more than one hour of deep isoflurane anesthesia. The primary effect of isoflurane is its immediate impact on Ca^{2+} activity and insulin secretion, hyperglycemia being a consequence of this impact. As demonstrated in Figure 4B, we present data from the same animal transitioning from the awake state to deep isoflurane anesthesia and back to the awake state, all within a one-hour timeframe. Given the short duration, hyperglycemia is unlikely to account for the pronounced suppression of Ca^{2+} activity during anesthesia under otherwise identical imaging conditions. To demonstrate the metabolic impact of isoflurane on pancreatic islet function, we included physiological data in Figure 4G, examining systemic glucose handling during a glucose tolerance test. These data reveal that the elevated blood glucose levels observed under isoflurane anesthesia result primarily from a near-complete suppression of insulin secretion.

This raises the important question to what extent can functional data such as β -cell Ca^{2+} activity still be considered physiologically relevant if the underlying metabolic function is severely impaired due to anesthesia? While we discuss potential mechanisms for this effect in the revised discussion section, we acknowledge that the exact pathways remain unclear. Our findings underscore the need for careful consideration of anesthesia-related confounding factors when investigating brain activity or, as in our case, excitable tissues such as pancreatic islets.

Revised Manuscript: Figure 4 now includes metabolic data on insulin secretion, and we have revised the Introduction and Discussion sections accordingly.

2. Figure 1G is really remarkable and it would be worth mentioning in the body of the text that this time composite image means that 10 minutes worth of recordings at 1 frame per second are superimposed here with no visible blur. Is this true for all images and is there a mathematical way of representing this eg by averaging displacement from the central (time course) image?

Thank you for encouraging us to strengthen this part of the manuscript. We have revised the relevant figure to better emphasize the stability aspect by now displaying GCaMP3 oscillations alongside the backscatter signal for selected ROIs corresponding to individual β -cells (Figure 1F/G). While we have not implemented a mathematical model, we hope that this revised presentation aligns with your expectations. In this analysis, we use $\Delta F/F$, where F represents the minimum fluorescence intensity value during the 10-minute recording period, rather than an averaged basal minimum. This approach allows us to present the raw signal dynamics as directly as possible, including the intrinsic noise level of our recordings.

Regarding signal stability, all "successful" recordings exhibited robust and stable signals. However, several practical factors can affect recording quality, including the precision of metal mount installation during surgery and microscopy, proper animal habituation to the imaging setup, and the avoidance of stress-induced panic behavior. Excessive movement may cause mechanical vibrations that shift the focal plane and introduce artifacts. To minimize these issues, all mounting components must be meticulously cleaned and assembled to ensure complete mechanical stability during imaging.

Revised Manuscript: Figure 1 has been updated to provide an improved representation of image stability by illustrating the backscatter signal over time.

3. Supplemental Figure 1 – again this is a reassuring dataset since stress-induced hyperglycaemia is an obvious concern here. Please add in a Supp 1b to show the

average (and range) blood sugar levels for each type of imaging session (eg awake vs immobilised) in absolute terms (mM or mg/dl).

We have included blood glucose measurements for each condition directly in the results section for Figure 4, taken after the respective 1–2-hour imaging sessions; awake (8.8 ± 0.5 mmol/L), isoflurane anesthesia (12.3 ± 0.8 mmol/L), and Hypnorm anesthesia (8.4 ± 0.8 mmol/L). These values indicate a moderate elevation in blood glucose under isoflurane anesthesia, which is consistent with previous findings by Akalestou et al. (Nature Communications, 2021, DOI: 10.1038/s41467-021-25423-8). In their study *“Intravital imaging of islet Ca^{2+} dynamics reveals enhanced β -cell connectivity after bariatric surgery in mice”*, they carefully reported blood glucose levels during imaging sessions under isoflurane, ranging from 12.5 ± 0.7 to 13.3 ± 0.5 mmol/L across different experiments. In this study, the underlying assumption is that moderate hyperglycemia under isoflurane does not constitute a confounding factor for Ca^{2+} imaging, provided that glucose levels are matched across all experimental conditions.

Revised Manuscript: We have included blood glucose post-measurements for each physiological condition in the results section for Figure 4.

4. Line 205: “Vascularization and innervation of pancreatic islets at the brain surface”
“The brain-transplanted pancreatic islets integrated well and rapidly on the brain surface and showed excellent survival when placed centrally not too close to the border of the cranial window.”

Either here or in methods please clarify – were you attempting to implant only one islet per brain window? What was the implantation success rate?

We apologize for the previous lack of clarity and have now provided a more precise description in the *Materials and Methods* section. Our approach involves attempting to place up to 10 pancreatic islets into the cranial window using Ringer’s solution. A

common technical challenge during this process is the displacement of unfixed islets caused by fluid movement and pressure changes when applying the glass coverslip, which can push islets toward the edge of the cranial window or out of view.

In the first week post-transplantation, it is not uncommon for islets located near the bone margin or fracture to disappear from view. We speculate that these islets may migrate or become displaced beneath the skull, rather than undergoing cell death. Importantly, we have never observed signs of apoptotic or necrotic islet loss, which would typically present as a gradual process over several days or weeks. In contrast, islets located within the central area of the cranial window, visible after one week, consistently engraft and remain viable.

To mitigate the issue of islet displacement during mounting, especially when transplanting larger quantities such as 50 human islets, we introduced a technical improvement using 2.5% gelatin in Ringer's solution. This modification effectively stabilizes the islets during the placement of the coverslip, ensuring their retention within the cranial window. We now mention this optimization in the revised manuscript.

Revised Manuscript: We have revised the Material/Method and Discussion sections accordingly.

5. Figure 2A-G - % islet volume covered with blood vessels seems an unusual way of describing the implantation process. There is insufficient data here to convince the reader that the blood vessels are extending throughout (growing into not simply overlaying) the islet and that their calibre/density is redolent of the native pancreas. In the Conclusion some discussion should be made about the blood supply of the dura (presumably the meningeal arteries, the venous drainage as well as the origin of the nerve supply).

We acknowledge the reviewer's critique and have improved both the data presentation and the accompanying explanation regarding islet vascularization. To clarify, the

vascularization image shown at 3 weeks in Figure 2A represents a single confocal section, approximately 30% into the islet volume, and not a maximum intensity projection. The TMR-labeled capillaries can be clearly visualized within the β -cell mass (GCaMP3 signal) as recessed areas, indicating the presence of intra-islet vasculature occupying space within the islet structure.

In our volumetric analysis using Velocity software, we define the GCaMP3 signal from z-stack recordings as representing the β -cell volume (Figure 2C). Within this defined volume, we calculate the TMR-labeled vessel signal as a percentage to quantify intra-islet vascularization. To further support this, we have now included a 3D volume rendering with orthogonal (xy, yz, xz) views in the revised Figure 2B. This provides a clear spatial illustration confirming that the vessels are located within the islet and not merely on its surface.

Regarding vascular origin, we hypothesize that the blood supply is derived from the dura mater overlying the somatosensory cortex, specifically from meningeal vessels such as fine arterioles. We have expanded the discussion to address neural innervation, comparing the native neural connectivity of pancreatic islets with that observed in dura mater–engrafted islets. We also consider the possibility that these islets receive innervation from a distinct set of ganglia compared to native islets.

Revised Manuscript: We have revised the Discussion sections accordingly and added volumetric representation in Figure 2.

6. The data implying innervation is basic in the sense that there is no attempt to compare the density or distribution of the innervation pattern with respect to this group's well established eye model or the endogenous pancreas. For example the TH and VAChT images look to be peripheral and sparse. Given that some of the dural innervation comes from the vagus, this is a missed opportunity to show that this model may well have strengths in terms of the origin of the innervation. This would be an

important study in the future, in order to cement this platform as a good model for studying islets.

We fully agree that investigating the innervation pattern and functional neural connectivity of dura mater–engrafted islets is highly important and represents an exciting avenue for future research. However, we consider this aspect beyond the scope of the current study.

To clarify our position, we primarily compare our findings to our established intraocular islet transplantation model and previously published data using identical antibody staining protocols and cryosectioning techniques. Specifically, we refer to the work by Rodríguez-Díaz et al. ("*Noninvasive in vivo model demonstrating the effects of autonomic innervation on pancreatic islet function*"). Proc. Natl. Acad. Sci. U. S. A. 109, 21456–21461 (2012). DOI: 10.1073/pnas.1211659110), who examined intraocular islet innervation in comparison to native pancreatic islets. We kindly ask you to pay particular attention to Figure 1 in this publication. Using the same methodology, our current data suggest that islet grafts under the cranial window do not exhibit hyperinnervation, a valid concern given the proximity to the brain, but rather display an innervation pattern similar to that seen in the eye model.

However, as we acknowledge in the Discussion section, our analysis does not include detailed anatomical quantification or functional assessment of neural connectivity, which remains a limitation of the current study.

For your information, we did attempt whole-mount immunostaining of recovered islet grafts and evaluated multiple tissue-clearing protocols. Unfortunately, these efforts were unsuccessful, likely due to the high optical density of pancreatic islets, which results from the accumulation of insulin crystals in densely packed secretory vesicles within β -cells. Moving forward, transgenic reporter mouse models with endogenous expression of neural markers will be essential to better characterize the neuroanatomy of islet grafts in this context.

Revised Manuscript: We have revised the Discussion sections accordingly and added another TH-positive immunostaining of dura mater-engrafted islets in Figure 2F.

7. **Blood flow videos:** supp video 1 and 2 are BOTH from awake animals?

Yes, both recordings were acquired from the same awake mouse. One of them is shown together with the backscatter signal, which serves as an indicator of the islet graft volume.

Why did you not provide a comparator from anaesthetised experiment?

Agreed. We have included representative red blood cell (RBC) flow recordings from both conditions to support our findings (Supplementary Movie 1-4).

Were these velocity experiments performed on a single animal once awake and once under anaesthesia?

That is correct. To minimize variability from external factors, the comparison was performed within the same animal. First, RBC velocity was recorded in the awake state following the injection of labeled red blood cells. After a 2-hour rest period, a second recording was obtained under isoflurane-induced anesthesia, all on the same day. This experimental design ensures a direct within-subject comparison, and the observed effect on RBC velocity is reproducible across individual animals.

Without this info it's difficult for me to comment more at this stage however:

7.1. These images are extremely impressive and I commend the experimentalists for their work.

We sincerely thank the reviewer for this kind comment and greatly appreciate the recognition of the experimental effort involved.

7.2. Isoflurane is well documented to increase cortical blood flow velocity so this at first glance makes sense. However, the animals were obviously deeply anaesthetised for an hour. This means they were also likely hypercapnoeic (a confounder) and this doesn't represent the depth or duration of anaesthesia for other islet imaging experiments eg in the eye.

These are indeed important physiological considerations. In the revised manuscript, we have expanded our discussion to relate our findings to existing literature, highlighting the known isoflurane-induced increase in cortical blood flow velocity as a likely contributing factor to our observations.

However, we would like to note that we consider hypercapnia an unlikely confounder in our experimental setup for two key reasons. First, the RBC velocity recordings were performed within a controlled time window of approximately 10–30 minutes following the induction of isoflurane anesthesia, rather than after prolonged exposure. Second, we take great care to minimize CO₂ accumulation in our subjects by precisely adjusting the oxygen–isoflurane mixture to a concentration between 1.0% and 1.5%, carefully titrated according to the animal's breathing pattern. Our goal is to avoid both gasping ("snap-breathing") and rapid shallow breathing, which is standard practice for all of our intravital imaging and surgical procedures involving isoflurane.

7.3. If blood vessel calibre is not changed, why is RBC flow faster? I see that you have tried to conclude on this in the Discussion but I think that the conclusions made there are too bold and no consideration of the myriad of other confounders eg depth of anaesthesia have been considered.

We do not know the exact mechanism and can only speculate, as we have not conducted further experiments to clarify this question. We acknowledge that we missed the opportunity to assess whether pericytes play a confounding role in this context. Pericytes in islet capillaries are known to contract or relax in response to various signals, including neurotransmitters, glucose, and hormones like insulin, thereby modulating capillary diameter and local blood flow. Importantly, pericytes do not induce widespread dilation or constriction of entire capillaries. Instead, they regulate localized flow control by adjusting bottleneck-like regions along the capillary, particularly at branch points or constricted segments.

Joana Almaca and Alejandro Caicedo published an insightful study in *Nature Communications* (2018), titled "*The Pericyte of the Pancreatic Islet Regulates Capillary Diameter and Local Blood Flow* (Cell Metab. 27, 630-644.e4 (2018))," which provides key evidence on this topic.

Given that islet capillaries are small and lack smooth muscle, unlike arterioles, global diameter changes are unlikely, and the pericyte-mediated mechanisms appear neither prominent nor relevant here. Instead, we attribute the increased blood flow velocity primarily to the isoflurane-induced elevation in cortical blood flow, suggesting this is a site-specific effect.

Revised Manuscript: In the sections Materials/Methods and Discussion, we have revised the part on blood flow velocity accordingly.

8. Calcium imaging section

8.1. Supplemental video 3 is a fantastic achievement.

Visual assessment of this video looks different to calcium waves that have previously been imaged in vitro and in vivo. There is no obvious starting zone/leader area or "wavefront" and I wonder if this is because of the speed of the GCaMP3 and your frame rate – please comment (work of Paukert springs to mind comparing GCaMP 3 and 6 in being able to distinguish neuronal wave patterns).

Thank you for this thoughtful feedback. We agree this is an important point and it prompted us to critically reevaluate our approach to analyzing β -cell Ca^{2+} dynamics. As the reviewer rightly notes, studies such as Paukert et al. (*PLOS ONE*, 2017, "Comparison of GCaMP3 and GCaMP6f for studying astrocyte Ca^{2+} dynamics in the awake mouse brain.") have shown that GCaMP3 performs well for detecting robust, global Ca^{2+} signals, whereas more recent indicators like GCaMP6f offer improved sensitivity and temporal resolution for fast or localized events. Similarly, *in vivo* studies describing leader cells (e.g., Salem et al., *Nat Metab*, 2019, "Leader β -cells coordinate Ca^{2+} dynamics across pancreatic islets *in vivo*.") employed GCaMP6s, semi-3D imaging, and faster acquisition rates (3 Hz), which are better suited to capture the complexity of β -cell activity across space and time.

In our initial analyses, we explored several advanced parameters using our MATLAB-based pipeline, including fast Ca^{2+} oscillations (6–60 s), first responder cell identification, and spatial patterns such as zones and hubs. However, we encountered inconsistencies in detecting fast or fine-scale events, which we traced back to fundamental limitations of our imaging system. Our confocal setup is limited to a single focal plane and a maximum acquisition speed of 1 Hz. Attempts to increase frame rate lead to reduced spatial resolution and a restricted field of view, preventing complete islet coverage. While our calcium indicator, GCaMP3, provides robust and stable data in our established intraocular islet model (Ins1Cre-GCaMP3), it lacks the temporal resolution of GCaMP6 variants and is therefore potentially less suited for capturing rapid Ca^{2+} dynamics.

In light of these limitations, we have revised our manuscript to focus on parameters that are robust and consistently measurable within our system such as slow Ca^{2+} oscillation amplitude, period, correlation, and plateau fraction. These metrics reliably reflect islet-wide activity and are not confounded by the technical constraints of our setup or biological variability in the data. We appreciate the reviewer's comment, which helped us refine the scientific focus of the manuscript and ensure that our conclusions are well-supported by our methodology.

Revised Manuscript: We have completely revised our Ca^{2+} data analysis and amended Figures and Manuscript accordingly.

8.2. I agree that amplitude decreases with anaesthesia. I'm not sure I agree with your conclusions about periodicity (which is not defined in your methods). What is happening is that a few of the beta cells that were already dark (cells at the bottom of your image 4A) fail to reach your threshold for activation and therefore become "disconnected" – the Pearson heatmaps show very clearly that the same cells (in an offset cross pattern) that are already poorly connected in the awake setting become darker blue when anaesthetised because of dampened beta cell activity globally. I therefore find the connectivity analyses here unhelpful in describing what you are observing. Fundamentally the pattern of connectivity is preserved it's just that the entire signal is dampened.

We agree that signal dampening during transitions between awake and isoflurane-anesthetized states, and vice versa, may contribute to reduced synchrony within the β -cell network. However, in the imaging sessions where each anesthesia condition was recorded independently, this dampening effect does not account for the observed differences in signal coordination. In these cases, imaging settings were optimized separately for each session, adjusting for the specific fluorescence intensities under each condition and thereby minimizing potential artifacts due to signal attenuation. We have revisited our Ca^{2+} data and chosen a different representation compared to the initial manuscript. We now aggregate the single-cell data of separate recordings prior to representation and perform statistical analysis on the combined dataset. While we continue to observe a modest reduction in overall β -cell network connectivity under isoflurane, this difference does not reach statistical significance (Figure 4I). In the revised analysis, we now perform comparisons at the islet level, which better reflects the structure of individual experiments and accounts for biological variability across

islets. This approach yields more stringent and representative statistical comparisons. Using this method, differences in Ca^{2+} amplitude between conditions remain statistically significant, while periodicity differences no longer reach significance (Figure 4H). To enhance clarity, we have now expanded the Methods section to include a more detailed explanation of how periodicity is quantified.

Revised Manuscript: We have completely revised our Ca^{2+} data analysis, including statistical calculation and amended Figures and Manuscript accordingly.

8.3. For the experiment comparing different anaesthetic agents – you mention the “same islet grafts” were repeatedly imaged – please clarify how many different islets/animals? The cross section that you choose to display does not look identical in the three different conditions (isoflurane one looks different).

The same islet was recorded once under each condition. *N* indicates the number of individual islets and represents independent recording sessions. The islets were transplanted into four recipient animals in total. We have now clarified this in the figure legends to avoid confusion.

The representative images shown are from the same islet graft. It is correct that the imaged area may appear slightly different under isoflurane compared to hypnorm and awake conditions. These visual differences are due to the fact that imaging under both anesthetics cannot be performed on the same day. As a result, the focal plane and imaged area may differ slightly, particularly when a different operator handles the setup. The awake and hypnorm images appear more similar because, although the animal was completely removed from the microscope setup, the stage position parameters allowed for nearly identical repositioning. Isoflurane imaging was performed one week later to allow full recovery from the deep hypnorm anesthesia.

8.4. Please define how you analysed wave parameters such as amplitude and “period” in methods. It also looks like your duty cycle (probably a better measure of insulin secretion and roughly the product between frequency and duration) is altered in anaesthesia. For a broader audience it would also be helpful to define what you mean by slow and rapid oscillations in your methods. For example in Fig 4b there clearly are fast oscillations superimposed on the slower waves in the isofluorance condition but I’m not sure if you are reporting them as I do not know what your cut off is for defining a “fast wave”.

A more detailed explanation of how amplitude and periodicity are analyzed is now included in the Methods section. In addition to these two parameters, we have incorporated the plateau fraction, which measures the proportion of time a cell remains in its active state (as described in Satin et al., 2015, “*Pulsatile insulin secretion, impaired glucose tolerance, and type 2 diabetes*”, *Mol. Aspects Med.* 42, 61–77). This parameter shows a clear decrease under isoflurane anesthesia, indicating a reduced duration of β -cell activity compared to the awake condition.

In the previous version of the manuscript, we defined fast oscillations as those with a period of 6 to 60 seconds, and slow oscillations as those lasting between 60 and 600 seconds. In our model, and under the current experimental conditions, slow oscillations predominate. Therefore, in this revised version, we have chosen to focus exclusively on slow oscillations. This decision also reflects the technical limitations of our setup in reliably detecting fast oscillations, given the use of GCaMP3-expressing islets (which have limited sensitivity and temporal resolution) and a confocal imaging system with insufficient frame rate to accurately capture rapid Ca^{2+} dynamics.

Revised Manuscript: We have revised the part on Ca^{2+} data analysis in the sections Materials/Methods accordingly.

8.5. Line 322 “Compared to non-anesthetised mice, the degree of synchronizaton deviate more under isoflurane anesthesia than under administration with Hypnorm” I disagree with this. Firstly your “exemplar” comparisons clearly shows that hypnorm results in a much poorer average R value – much more blue! The combined data in 4J is not conclusive and when analysed using an ANOVA with post hoc correction the blue and orange bars are NOT different.

As stated in point 8.2, we have now analyzed differences between islets within each group, rather than pooling all individual cells from different islets. Using a one-way ANOVA followed by Tukey post hoc test, we found no statistically significance for altered network synchronization when comparing isoflurane to awake conditions and Hypnorm (Figure 4I).

8.6. The attempt to define leaders is interesting but the data too weak to draw conclusions from. There is no attempt to quantify “shift” in terms of their identity or position related to baseline/non anaesthetised. Also, without comparator data from the same length of imaging in animals awake the whole time it is impossible to tell whether this shift wouldn’t have happened anyway. In fact, as supp Fig 4 shows, the zonal location of your first responders is actually very well preserved throughout anaesthesia.

Granger – given that you have already concluded that isoflurane reduces the number of connected cells it’s hardly surprising that the Granger associations are also reduced. I don’t think this adds much I’m afraid.

We agree that, based on the current data, it is difficult to draw any definitive conclusions regarding leader cells. As pointed out in 8.1, we became aware of our technical limitations. Therefore, we have decided to remove this analysis from the manuscript and instead focus on islet physiology and the characterization of the dura

mater transplantation site, specifically comparing Ca^{2+} dynamics under non-glucose-stimulated conditions and following glucose challenge.

9.DISCUSSION

9.1. This is the first mention of meningeal puncture to allow CSF leak to bathe the islets – if this is central to the success/survival of the transplants then it needs to be very clearly described in the surgical Methods (presumably a single puncture hole with a 27G needle)

We appreciate the reviewer's attention to the surgical details. The procedure involves a single dural puncture using two 27G needles: one hand gently lifts a slightly elevated area of the dura mater, while the other hand applies a controlled puncture against that raised point. It is reasonable to assume that, beyond this deliberate intervention, the dura mater may also experience minor microincisions during the drilling process and skull plate removal. Nevertheless, we included this dural puncture step systematically to ensure consistent cerebrospinal fluid (CSF) entry into the cranial window and to standardize conditions across preparations. This procedural detail has now been explicitly described in the Materials and Methods section of the revised manuscript.

Revised Manuscript: We have revised the Material/Methods sections accordingly.

9.2. This model is being posited as a platform for tissue modelling without anaesthesia. This is very exciting but there is no discussion as to whether other transplanted tissue types have been attempted or if there is evidence that they are likely to engraft?

We thank the reviewer for this important comment. We agree and have revised our manuscript to tone down any broad claims regarding the dura mater as a general transplantation site, as we currently lack sufficient experimental evidence to support

such a statement. However, our revised manuscript now demonstrates successful interspecies transplantation of human pancreatic islets to the dura mater.

Revised Manuscript: We have revised the Discussion sections accordingly.

9.3. Were you able to detect calcium oscillations in every islet in every imaging attempt? How often were you able to image calcium oscillations in all elements of the imaging protocol (pre, induction, anaesthetised, recovery?) The data is presented in such a way that this is always measurable – is that true?

Yes, this is largely accurate. Beginning approximately 2–3 weeks post-transplantation, we consistently observed calcium oscillatory activity in the pancreatic islet grafts placed on the dura mater. Specifically, we detected slow Ca^{2+} oscillations under normal physiological blood glucose levels.

A lack of Ca^{2+} activity was only observed in exceptional cases such as in fasted animals with profoundly reduced blood glucose levels (<5 mmol/L) during either isoflurane anesthesia or awake imaging. These findings support the notion that the transplanted islets remain viable and functionally responsive under physiological conditions, with Ca^{2+} dynamics being tightly linked to the systemic metabolic state.

9.4. I think it might be worth noting that overall patterns of connectivity do seem to be preserved with isoflurane imaging and that the comparison of different islet stimuli that are imaged and compared under isoflurane will still have relevance, latest caveats accepted.

We agree with the reviewer that the reduced correlation of β -cell network activity can be attributed to a general reduction in Ca^{2+} activity or a shift of the signal outside the optimal detection range. This explanation is particularly relevant for the one-hour recordings performed during transient isoflurane anesthesia. In additional recordings

performed independently across different islets, animals, and days, we carefully adjusted the laser power to remain within an unsaturated and optimal detection range. Indeed, with the revised Ca^{2+} data analysis (see point 8.2), the correlation of β -cell activity was not significantly altered under isoflurane anesthesia. We would like to emphasize that it is not our intention to discredit the use of isoflurane in intravital imaging. Isoflurane-based protocols remain valuable, particularly when used consistently across control and experimental groups. Nonetheless, our data clearly indicate that isoflurane anesthesia can induce subtle but significant shifts in physiological Ca^{2+} signaling, especially in excitable tissues such as pancreatic β -cells. These potential effects should be carefully considered when designing and interpreting functional imaging experiments.

Revised Manuscript: We have moderated our critique of anesthesia protocols used in intravital imaging, adopting a more balanced tone, and have revised the Discussion sections accordingly.

Reviewer #2 (Remarks to the Author):

The aim of this study is to set up a new technique to image pancreatic islet cells in awake mice. To overcome the fact that the pancreas is anatomically located deep in the abdominal cavity, which presents as a significant difficulty on intravital microscopy, the authors transplanted the islet cells onto the surface of the brain to improve imaging accessibility. Additionally, the authors described the measurement of calcium dynamics in these transplanted cells, as well as blood flow velocity of the islet organoids, in awake mice with head fixation. There are many critical concerns with the methodology and experimental design that need to be addressed prior to publication:

We sincerely thank reviewer #2 for his/her time, thoughtful critique, and valuable scientific input. We particularly appreciate the fundamental questions raised regarding the methodology and the physiological characteristics of the cranial window model. We acknowledge that we initially overlooked some potential technical challenges associated with this approach, which we have now addressed in the revised manuscript. In addition, we have conducted further experiments to better characterize the physiological integration of dura mater–engrafted islets. In our view, the revision has substantially improved the quality and clarity of the study, and we hope that they satisfactorily address the reviewer's concerns.

Major:

1) The surgical removal of the skull to generate a transparent cranial window will create the bulging of the brain through the opening as a response from existing intracranial pressure. Arguably, this is very far from "non-invasive" as the titled suggested. As such, this term should be removed entirely from the manuscript. What are the implications on brain integrity, stability and inflammation?

We thank the reviewer for his/her insightful comments. We have addressed all points raised through appropriate revisions to the manuscript, including additional experiments or references.

Brain Bulging

We were able to prevent brain bulging in all animals. When the craniotomy is performed properly by experienced personnel, complications such as brain protrusion are typically avoided, particularly when the removed skull area remains below 20 mm². In our protocol, the craniotomy is limited to a maximum of 16 mm², which lies well within the described safe threshold. Nevertheless, we fully acknowledge that intracranial pressure can lead to brain protrusion following skull removal, a risk well recognized in cranial window preparations.

This method has been widely and successfully used for decades, with troubleshooting strategies comprehensively documented in the literature. Accordingly, we implemented multiple countermeasures to minimize risk, including maintaining dura mater surface moisture and temperature during drilling, applying minimal pressure to the skull during the craniotomy, minimizing dura exposure time, and ensuring appropriate anesthesia depth. These precautions allowed us to consistently avoid complications associated with brain bulging. We note that peripheral administration of dexamethasone, often used to reduce brain swelling, was not necessary in our study. These considerations and corresponding references (Cramer et al., *J. Neurosci. Methods*, 2021 (DOI: 10.1016/j.jneumeth.2021.109100); Holtmaat et al., *Nat. Protoc.*, 2009 (DOI: <https://doi.org/10.1038/nprot.2009.89>)) are now explicitly addressed in the revised methods section. We emphasize that correct surgical execution is critical for both animal welfare and the quality of subsequent intravital imaging.

Terminology: “Non-Invasive”

We agree with the reviewer that the term "non-invasive" may be misleading in this context and have removed it from the title and revised its usage throughout the text.

Our original intent was to distinguish the longitudinal, repeated intravital imaging under awake conditions (which involves no further physical manipulation after cranial window installation) from other methods requiring repeated or deeper surgical intervention, such as the insertion of GRIN lenses or prisms into the brain. However, we acknowledge that the term may imply the absence of any surgical procedure, which is not accurate, and have clarified our terminology accordingly.

Brain Integrity, Stability, and Inflammation

Throughout the study, we observed no signs of compromised brain integrity, instability, or inflammation. To further support this, we have now included data on three systemic inflammatory markers, analyzed during the first weeks following surgery and transplantation, which showed no significant elevation (Figure 1E). These findings are consistent with existing literature on immune responses in cranial window models.

We also address this issue more explicitly in the revised discussion. Importantly, no antibiotics were administered at any point, and all animals underwent extensive health screening during a preliminary study prior to receiving ethical approval. Surgical procedures were carefully refined during this phase to ensure optimal outcomes.

Animals remained in excellent health throughout the experimental timeline. All exhibited clear cranial windows and stable physiological conditions. The average welfare score was 0.4 on the KI-mallen scale (used in the Swedish Animal Welfare Act to assess humane endpoints), placing them only marginally above the baseline score of unmanipulated animals. Furthermore, none of the experimental animals showed behavioral signs indicative of inflammation or neurological compromise (e.g. hyperactivity, lethargy, ataxia, tremors, seizures, abnormal nesting, circling, head tilt, anorexia, excessive grooming, or self-injury). All animals were regularly monitored by independent veterinarians who confirmed their welfare status.

Revised Manuscript: We have revised the Material/Methods and Discussion sections accordingly. Figure 1E shows the inflammatory marker after cranial surgery and islet transplantation.

2) The transplanted organoid is placed onto the dura. How does this impact on the perfusion of the dura? Moreover, what is the effect of the cerebral spinal fluid on the 'micro-organs', or in this case, the transplanted islets versus a naïve pancreas?

We argue that the mechanical load imposed by islet transplantation to the dura mater is minimal. Each pancreatic islet (diameter ~150–250 μm ; volume <0.3 μL) contributes only a small tissue volume, even when 10–50 islets are transplanted, amounting to just a few μL in total. Our empirical observations indicate that this mass is well below the threshold required to compromise dural perfusion or alter intracranial pressure. Consistent with this, our data demonstrate successful angiogenic integration. Within the first two weeks, penetrating dural microvessels invaded the graft, and by week 3, a substantial portion of the β -cell volume was perfused by TMR-labelled capillaries (Figure 2A). This indicates that the graft becomes integrated into, rather than obstructive to, the host dural microcirculation.

Regarding exposure to cerebrospinal fluid (CSF), we note that this exposure is both transient and external. The islets are positioned on the outer surface of an intact dura mater. CSF enters the cranial window through a 27 G relief puncture and may transiently bathe the graft externally. However, following vascularization, the metabolic environment of the islets is dominated by systemic blood supply rather than CSF.

3) On the same train of thought, the authors demonstrate positive staining for enzymes for parasympathetic and sympathetic neurons in transplanted cells. How is the ratio/composition compared to the pancreatic ones in situ? More importantly, are these neurons functional?

We appreciate the reviewer's insightful question regarding the neuroanatomical integration of the islet grafts. In our study, we employed the same antibodies and comparable staining protocols as those used by Rodríguez-Díaz et al. ("*Noninvasive in vivo model demonstrating the effects of autonomic innervation on pancreatic islet function*"). Proc. Natl. Acad. Sci. U. S. A. 109, 21456–21461 (2012). DOI: 10.1073/pnas.1211659110), who thoroughly characterized the neuroanatomical distribution of nerve fibers in native and intraocular transplanted pancreatic islets. We kindly ask the reviewer to pay particular attention to Figure 1 in this publication. We discuss these parallels in detail in our revised manuscript. While our current study does not include a comprehensive quantitative analysis of innervation, our data suggest that islet grafts on the dura mater possess the structural potential to receive physiological neuronal input, comparable to what has been observed in the intraocular model. Importantly, we found no evidence of hyperinnervation, a valid concern for grafts placed near the brain.

That said, we acknowledge the absence of direct functional proof of neuronal connectivity, which we clearly define as beyond the scope of the present study. Future investigations using transgenic reporter mouse lines or optogenetic approaches will be necessary to directly assess this.

Nevertheless, we provide indirect functional evidence of neural and systemic integration. In newly added experiments, human pancreatic islet grafts on the dura mater displayed appropriate metabolic responses under fed, fasted, and glucose-stimulated conditions. These responses mirror those seen in native pancreatic islets and are indicative of intact physiological regulation. Given that such regulation is multifactorial and includes neural, hormonal, and metabolic inputs, we argue that the observed functionality suggests that neuronal input, if present, is at least not impaired or obstructive in our model.

Revised Manuscript: We have revised the Discussion sections accordingly and added another TH-positive immunostaining of dura mater-engrafted islets in Figure 2F.

4) Successful islet perfusion was shown, but how about endothelium activation/integrity markers? Does the transcriptional signature of the islet vasculature match that of the brain, or the pancreas?

We fully agree that this is a crucial point, as vascular integrity is a central component of the hormone-secretory function of pancreatic islets. To address this, we conducted vascular permeability experiments using both low and high molecular weight dextrans. These experiments allowed us to quantify the permeability index of vessels within the dura mater and the intra-islet vasculature, revealing clear differences in permeability (Figure 3A-D).

This functional assessments indicates that the islet microenvironment actively contributes to the specification of the capillary phenotype. Our findings are consistent with prior literature demonstrating that VEGF-A, secreted by islet tissue, plays a pivotal role in shaping local vascular characteristics.

We have expanded on these findings in the revised manuscript. However, in response to the reviewer's suggestion, we would like to clarify that a comprehensive analysis of the transcriptional signature of the engrafted vasculature is technically beyond the scope of the present study. That said, we agree that this is a valuable and promising direction for future investigations aimed at understanding the molecular determinants of graft vascularization.

Revised Manuscript: We have revised the Discussion sections accordingly and included a new experiment assessing vascular permeability, presented in Figure 3.

5) Whilst the brain has an anatomical advantage that allowed the authors to imaging this tissue bed when the animal is awake, the study design ignored the biological implication on this very complicated and rather unique organ. The brain itself does not

harbour any pain receptors which is an advantage for transplantation studies, however the removal of the skull and the associated surgical manipulation will induce pain. Was any analgesia used beyond day 3 post-surgery? The administration of analgesia will impact on inflammation and subsequently blood flow.

We appreciate the reviewer's concern and would like to clarify several important points. The islet grafts are placed on the dura mater, which is densely innervated by trigeminal sensory afferents, and thus inherently sensitive to nociceptive stimuli. This anatomical context is well established in headache research. To manage post-operative pain, we used buprenorphine (administered twice daily for 3 days), following veterinary recommendations (outlined in the Materials and Methods section). Buprenorphine is effective without significantly affecting coagulation or wound healing, unlike NSAIDs. Importantly, animals showed no signs of persistent pain beyond the buprenorphine treatment window, as confirmed by routine monitoring by animal care staff.

6) Furthermore, organoid/cell transplantation in immune-competent animals will generate a host response that is highly inflammatory which impact on blood flow. These factors need to be considered and measured prior, during and after to transplantation/imaging.

The reviewer is correct, this is of course an important concern that must be carefully considered in any transplantation study. However, the risk of immune rejection applies primarily to allogeneic transplantations (e.g., Balb/c islets into C57BL/6J hosts). In our study, we performed exclusively syngeneic transplantations, in which both donor and recipient animals share an identical and continuously maintained genetic background through regular backcrossing of the transgenic lines. This is a standard approach in murine transplantation models to avoid immune responses associated with MHC mismatches. For the human islet transplantation, we used immunodeficient Rag1^{-/-} mice to prevent xenogeneic immune rejection, which is also a widely accepted method

in such models. Furthermore, regarding potential inflammation we refer the reviewer to point 1 above.

7) The authors were very keen to showcase their analysis of calcium oscillation, however calcium signalling suggests the cell is alive but not necessarily functional. The insulin production/cell should be shown from the islet vs an islet in the pancreas in situ.

We fully agree with the reviewer that this is a key physiological aspect that was previously missing and left the relevance of β -cell Ca^{2+} dynamics open to interpretation regarding functional metabolic integration. We have now addressed this by demonstrating that the transplanted islet grafts exhibit full metabolic functionality, with insulin secretory responses comparable to native pancreatic islets (see revised Figure 5). To do so, we transplanted human pancreatic islets onto the dura mater, which enabled us to distinguish between human C-peptide (co-secreted with insulin in equimolar amounts) and endogenous mouse insulin. This allowed us to directly assess graft-specific insulin secretion in response to glucose stimulation.

These new data now allow us to relate the observed β -cell Ca^{2+} dynamics to functional insulin secretion, demonstrating that the transplanted islets are not only viable and vascularized, but also metabolically integrated and functionally competent within the host environment.

Revised Manuscript: We have included a new experiment assessing C-peptide secretion of dura mater-engrafted human islets, presented in Figure 5.

8) Similarly, the study design does not allow for biological-relevance comparison. What happens to the islet perfusion and their function when mice are eating, or recently had a meal? What happens when the mice have an administration of sugary drinks?

We appreciate the reviewer's valuable suggestion and have taken steps to address it by characterizing the Ca^{2+} response of the islet grafts following subcutaneous glucose administration, providing a biologically relevant functional comparison (Figure 6).

While we have also explored more naturalistic stimulation paradigms such as sugar-enriched drinking water or food intake, these approaches presented significant challenges. In particular, oral administration via sugary drinks led to high variability in glycemic response, and without the ability to continuously monitor blood glucose in awake, freely moving animals, we found it difficult to precisely correlate metabolic demand with the observed Ca^{2+} dynamics. For the current study, we have deliberately refrained from implementing these approaches due to the lack of precision and reproducibility they presently offer.

Revised Manuscript: We have included a new experiment assessing the effect of subcutaneous glucose administration on Ca^{2+} activity in dura mater-engrafted islets, presented in Figure 6.

9) The authors are making us aware that isoflurane will inevitably cause hyperglycaemia, but how biologically relevant is this (especially if imaging is not performed chronically)? In other words, assuming the imaging is done acutely on multiple timepoints within the same animal, does the blood glucose fluctuate to a level that is considered pathogenic?

We acknowledge that the previous version of the manuscript, particularly the Introduction, may have overemphasized the issue of hyperglycemia. In the revised version, we have moderated our language and reframed the physiological relevance accordingly. Elevated blood glucose is indeed a critical parameter in studies of β -cell Ca^{2+} dynamics and secretory activity, as it is the primary trigger for intracellular signaling cascades in β -cells. While moderate elevations, such as the 3–4 mmol/L

increase we consistently observe following isoflurane anesthesia, are not pathological per se, they must be considered when interpreting β -cell function.

Importantly, our revised manuscript now includes revised Figure 4G, which demonstrates that under isoflurane anesthesia, insulin secretion in response to glucose stimulation is dramatically suppressed. This suppression is accompanied by marked changes in both the intensity and plateau fraction of β -cell Ca^{2+} signaling. These findings highlight that isoflurane does not merely alter baseline glucose levels but interferes with the core β -cell stimulus–secretion coupling mechanism. This central impairment underscores a critical physiological divergence from the awake state.

Revised Manuscript: We have moderated our critique of anesthesia protocols used in intravital imaging, adopting a more balanced tone, and have revised the Discussion sections accordingly.

Minor:

1) Fig 2 A-E is not referred to in text.

We apologize for the oversight and have now ensured that every figure is clearly and appropriately referenced in the revised manuscript.

2) Figure 2 A is a single Z-stack, which seemingly shows the dura vasculature, and does not confirm vascularisation of the islet. Multiple z-stacks should be shown here and 3-D reconstituted to shown integration of vasculature into the islet.

While we respectfully disagree with the concern that Figure 2A depicts dura mater vasculature, we acknowledge the reviewer's point. The z-stack, extending approximately 30% into the β -cell mass volume, clearly shows cellular gaps in the GCaMP3 signal that align precisely with the TMR-labeled vasculature. We interpret this as intra-islet vasculature. Nonetheless, we appreciate the suggestion and have added

multi-stack reconstructions and 3D volumetric representations in Figure 2B to unequivocally confirm the intra-islet vascularization.

3) Fig 2E – what is 100% vascularisation? What is the score 'relative' to?

We have removed the previous representation of revascularization dynamics, where 100% was defined as the vessel volume relative to total β -cell volume at 10 weeks. Instead, we now present vascularization dynamics only as vessel volume normalized to β -cell volume, based on analysis using Velocity software.

4) Fig 3 – heart rate, blood pressure and full blood counts should be shown here.

Agreed, those parameters would indeed provide valuable insight. However, during recordings in awake, freely moving animals, applying additional monitoring devices is technically challenging and introduces stress or artifacts. While heart rate and blood pressure can be measured using non-invasive methods such as telemetry or tail-cuff systems, these require prior implantation or restraint, which can affect the animal's behavior and physiological state during imaging. Similarly, obtaining full blood counts would involve blood sampling, which is invasive and not feasible during imaging sessions. Therefore, in our current setup, continuous monitoring of these parameters was not performed but could be considered for future studies with appropriate adaptations.

5) Discussion is not written at an adequate quality, it does not discuss how the study integrate with existing literature.

We have completely revised the discussion and hope it now addresses the thorough concerns by the reviewer.

6) Line 407 - which study was being referred to? Their current study did not conduct a metabolic transplantation,... etc.

We appreciate the reviewer's careful reading and thoughtful comments. In our opinion, the study referenced in line 407 (Bloch et al., 2018, "*Intracranial Transplantation of Pancreatic Islets Attenuates Cognitive and Peripheral Metabolic Dysfunctions in a Rat Model of Sporadic Alzheimer's Disease.*" *J. Alzheimer's Dis.* 65, 1445–1458) involved intracranial transplantation of pancreatic islets aimed at improving metabolic and cognitive functions. While their primary focus included neurological outcomes, the transplantation of pancreatic islets inherently qualifies as a metabolic transplantation because it targets restoration of insulin secretion and metabolic regulation. Therefore, in our opinion, it is appropriate to classify their work within the scope of metabolic transplantation.

7) Line 420 - in comparison to the neuronal stains, at least the authors did compare how the blood flow velocity in their transplanted islets to the native ones. However, are the transplanted ones now perfusing too fast, assuming the *in situ* measurements quoted (500 $\mu\text{m}/\text{sec}$) were done with anaesthetics (ketamine). Their iso-treated blood flow was closer to 1000 $\mu\text{m}/\text{sec}$.

It is indeed challenging to establish definitive baseline values for intra-islet capillary flow velocity in fully awake, *in situ* pancreas due to technical limitations. The existing *in situ* data under ketamine anesthesia ($\sim 500 \mu\text{m}/\text{sec}$) likely reflect anesthetic-induced alterations. In our study, the awake imaging of transplanted islets shows flow velocities comparable to those reported under ketamine, suggesting our measurements approximate physiological conditions. The higher velocities observed under isoflurane anesthesia ($\sim 1000 \mu\text{m}/\text{sec}$) likely represent anesthetic-induced hyperemia rather than true physiological flow. Thus, while direct awake *in situ* pancreatic flow data remain

unavailable, our awake ectopic model offers a valuable approximation of physiological blood flow velocity.

8) Line 532 – a major overstatement saying all anaesthetic should be avoided during intravital microscopy. This needs to be rephrase and tone down to suggest appropriate controls are needed, and not completely remove anaesthesia.

We agree that the original statement overstated the point and have revised the wording to present a more balanced perspective. Rather than implying that all anesthetics should be categorically avoided during intravital microscopy, our aim is to highlight the importance of understanding the specific physiological effects anesthetics may introduce, particularly when studying excitable tissues such as pancreatic islets. Careful experimental design, including appropriate controls and condition-matched baselines, is essential and should be adapted to the specific scientific question and model system used.

Reviewer #3 (Remarks to the Author):

We thank Reviewer #3 for their thoughtful contribution to the review process and are grateful for his/her insights. We also appreciate Nature Communications' initiative to support and recognize Early Career Researchers through co-reviewing.

Reviewer #4 (Remarks to the Author):

In their paper, the Berggren lab describes a technique for intravital imaging brain-transplanted micro-organs in awake animals. There is a critical need in the field for intravital imaging of biological processes in awake animals, as anesthesia can potentially alter physiological states and research outcomes. The researchers have developed a novel microscopy platform that involves transplanting micro-organs, specifically pancreatic islets, onto the brains of mice, allowing for imaging of the tissues in awake animals.

While the method introduced is interesting, the paper could benefit from more explanations and demonstrations of applications. Additionally, some claims require further support. We have the following concerns/questions:

We sincerely thank Reviewer #4 for his/her thoughtful and constructive feedback. We carefully considered all points raised and have made substantial improvements to the manuscript, both experimentally and in the written presentation. We hope these revisions address the reviewer's concerns and meet his/her expectations.

Major Comments:

1. The paper explains well that imaging in awake animals is a significant step forward since current in vivo imaging requires animals to be anesthetized, which can alter tissue physiology and impact experimental outcomes. However, the paper does not adequately describe the disadvantages of their methods, specifically implanting organs in different niches. The brain is a unique organ with a very specific environment, which highly limits the experimental setting. Some biological conclusions that can only be drawn from physiological conditions (i.e., inside the actual organ within a specific niche) would still require the use of different imaging techniques, albeit with different types of anesthesia. In general, the paper focuses solely on the advantages of the newly developed technique and neglects to mention its limitations. This needs to be further

discussed in the introduction and/or discussion section of the manuscript. Furthermore, the paper would benefit from explicitly mentioning the specific applications of the method.

We agree that the original manuscript placed strong emphasis on the advantages of our approach. It was not our intention to downplay the value of established imaging systems, which are well-characterized and remain essential for studying tissue physiology *in situ*. Rather, our goal was to introduce a novel method that enables awake imaging of optically inaccessible tissues via a newly characterized transplantation site, offering unique opportunities for long-term, stable, and repetitive *in vivo* imaging. Nevertheless, the critique regarding altered microenvironments in transplantation studies is a general observation that applies to all ectopic transplantation models. A precise characterization is essential to biologically relate the new transplantation site to the *in situ* reference. To strengthen the physiological relevance of our approach, we have expanded the experimental characterization of the transplantation site and graft function. Specifically, we now include data on graft vasculature integrity (permeability assay, Figure 3A-D), metabolic integration of human islet grafts (Figure 5), and the functional response of grafts to systemic glucose stimulation (2-NBDG uptake and Ca²⁺ imaging, Figure 6). While our current focus is on pancreatic islets (both mouse and human), we acknowledge that evaluating the suitability of this transplantation site for other tissue types remains an open question and lies beyond the scope of the present study. We hope the revised manuscript now offers a more balanced and comprehensive perspective.

Revised Manuscript: We have now added more physiological characterization of the transplantation site, including a vascular permeability assay (Figure 3), assessment of islet graft metabolic function (Figure 5), and glucose-stimulated Ca²⁺ dynamics (Figure 6).

2. In line 127, the authors suggest that their technique can be used to image tissues/micro-organs/organoids implanted in the brain. Are these transplantation assays feasible? Can the authors cite other studies that have performed successful transplantations of other tissues in the brain recapitulating the physiological/pathological conditions?

We absolutely agree that this is an important aspect to address. For clarification, our transplantation model places pancreatic islets onto the dura mater, the outer meningeal layer of the brain. This approach minimizes invasiveness while maximizing optical access for *in vivo* imaging. To our knowledge, the dura mater has not been used as a transplantation site. In contrast, pancreatic islets have been extensively studied across various transplantation sites, making them an ideal reference tissue for establishing and validating a novel transplantation location. We describe this rationale clearly in the Introduction and provide a comprehensive discussion of the relevant literature. At the same time, we have moderated our claims regarding the general applicability of the dura mater as a transplantation site. We acknowledge the reviewer's valid point that, without additional experimental evidence, it is premature to assert that the dura mater is suitable for all tissue types.

Revised Manuscript: We have revised the Introduction and Discussion sections accordingly.

3. The paper claims broad applicability of the technique and differentiates it from existing methodologies. However, such a claim can only be made if they demonstrate multiple applications. For example, manipulating calcium dynamics with well-characterized drugs, such as an acetylcholinomimetic, could illustrate the versatility of the platform for functional assays and enhance the robustness of the authors' claims.

Alternatively, transplanting and imaging other tissues would support these claims. If this is not possible in a reasonable timeframe, it would be fair to tone down the claims of broad applicability.

We acknowledge the reviewer's valid concern and have revised the manuscript accordingly. Experimentally, we confirm that xenotransplantation of micro-organs, specifically human pancreatic islets, is feasible and that subcutaneous injection is a viable route for drug administration during Ca^{2+} imaging in awake animals. This demonstrates the potential for functional assays using pharmacological agents. However, as we have not yet demonstrated applicability to other tissue types, we have moderated our claims of broad applicability, highlighting this as a direction for future studies.

Revised Manuscript: Experimentally, we have now included human tissue transplantation (Figure 5) and characterized the subcutaneous glucose administration in awake mice during intravital imaging (Figure 6).

4. In Supplementary Figures 3 and 4, the authors characterize the first responder β -cells under anesthesia in comparison to the awake state. Do the data plotted in these figures originate from multiple mice? If not, can the authors comment on how reproducible these observations are across different mice and whether leader cell properties depend on the localization of the cell within the transplant?

The data in prior Supplementary Figures 3 and 4 represent a single one-hour recording illustrating the transition during transient isoflurane anesthesia in one experimental setup. Similar changes in first responder β -cell patterns were observed in other animals, indicating reproducibility. However, we have removed the first responder β -cell tracing from the revised manuscript. We acknowledge that our current imaging setup - limited

to a single focal plane and 1 Hz acquisition - lacks the spatial and temporal resolution necessary to robustly characterize such complex islet dynamics. To avoid overinterpretation, we now focus on more reliable Ca^{2+} measurements and note that future improvements in imaging resolution will be required to address these questions adequately.

Revised Manuscript: We have removed the former Supplementary Figures 3 and 4 from the revised manuscript.

Minor Comments:

1. The authors should consider rewriting the introduction section, as many aspects mentioned there are repetitive and do not add to the manuscript.

Thank you for your helpful comment. We agree with your assessment and have completely rewritten the introduction section. In the revised version, we have removed repetitive content and ensured that all included material directly supports the objectives and significance of the study. We believe this has improved the clarity and focus of the manuscript.

2. The authors mention that the time used for imaging the mice in their experimental setup is one hour. Can the imaging session be prolonged based on the scientific question, or is it restricted to short-term imaging? Can the same mouse be imaged multiple times (days, months)?

In principle, imaging sessions in our setup are not limited to one hour. They can be extended and repeated depending on the scientific objective. The primary limitations are ethical and related to animal welfare, rather than technical constraints.

Our ethical permit allows for up to 2 hours of awake imaging per day in the Mobile HomeCage, with a maximum of 5 consecutive imaging days followed by at least 5 days of rest. Long-term imaging is also permitted, with a total duration of up to 10 months. In this study, all mice were imaged multiple times, and some were followed for up to 6 months with viable islet grafts.

The system is specifically designed for longitudinal studies. We have for example monitored vascularization through a body window over a 10-week period in the same animal. Additional information about the Mobile HomeCage system is available at www.neurotar.com.

3. The authors should mention in the methods section how acclimatization is performed in more detail. Were the mice kept in the imaging holder for the same amount of time as the real imaging session on a daily basis? Were stress levels measured after the first imaging session when the mouse was awake before it underwent anesthesia?

We have added the requested details in the Methods section. The acclimatization protocol consisted of 5 consecutive days of one-hour daily sessions with the mice awake in the Mobile HomeCage, matching roughly the duration of the planned imaging sessions. This acclimatization began three weeks after transplantation, ensuring that awake imaging commenced four weeks post-transplantation (as illustrated in Figure 1D).

To assess stress adaptation, we evaluated a cohort of mice by measuring blood glucose levels before and after each training session. These data (presented in Supplementary Figure 1A) show a progressive reduction in glucose elevation over the acclimatization period, indicative of reduced stress responses. Given that glycemic control is a critical readout for islet graft function in our study, minimizing stress-related glucose fluctuations was important.

This protocol aligns with established practices for awake imaging using the Mobile HomeCage, as implemented by Neurotar (neurotar.com) and reported in previous studies examining sensitive physiological systems such as neuronal circuits.

4. The authors have shown that the pancreatic islet transplant is vascularized and innervated. Can they show, with whole mount staining, how this compares to the pancreas (organ) under normal homeostatic conditions? This kind of characterization is essential to determine the type of experiments that can be performed using this technique.

We fully agree that understanding the site-specific tissue integration is essential for interpreting data from ectopically transplanted tissues. In our study, we aimed to address this by combining quantitative analysis of revascularization and qualitative demonstration of neuroanatomical structure, alongside a comprehensive discussion of relevant literature. Numerous previous studies have characterized both native islets and islet grafts in ectopic sites such as the subcutaneous space, kidney capsule, and anterior chamber of the eye, particularly in terms of vascularization. These studies underscore the utility of pancreatic islets as a well-established and highly informative reference system for transplantation biology. In the revised manuscript, we have emphasized these comparative studies more clearly in the discussion. For example, we now highlight the work by Rodríguez-Díaz et al. ("*Noninvasive in vivo model demonstrating the effects of autonomic innervation on pancreatic islet function*"). Proc. Natl. Acad. Sci. U. S. A. 109, 21456–21461 (2012). DOI: 10.1073/pnas.1211659110), which offers valuable neuroanatomical comparisons between *in situ* and intraocular islets, suggesting that a similar level of neural integration may occur in our dura mater-engrafted islets. We kindly ask you to pay particular attention to Figure 1 in this publication. While we acknowledge that our current study lacks functional validation of neural connectivity, this limitation is explicitly stated in the discussion. Since immunohistochemical staining of innervation in whole-mount islets is technically

challenging and would require advanced clearing techniques or transgenic reporter models, we consider this beyond the scope of the current study, which focuses on functional and physiological integration.

To strengthen the manuscript, we expanded the functional characterization of the islet grafts, which serves as an integrative readout of inputs. As shown in Figure 3, we evaluate vascular integrity *in vivo*, and in Figure 5, we provide data on metabolic function, including glucose-stimulated C-peptide secretion from human islets engrafted on the dura mater. Importantly, these functional responses are benchmarked against native mouse islets, demonstrating that the grafts are not only structurally integrated, but also metabolically responsive to physiological stimuli, including fasting, feeding, and glucose challenge, which is comparable to pancreatic islets in their native environment.

5. In Figures 4g-j, the authors describe how oscillation periods and amplitudes differ between awake and anesthesia treatment. Can they highlight the profile of leader cells in these graphs?

We appreciate the reviewer's interest in the analysis of leader (or first responder) cells. However, as noted previously, we decided to focus our Ca^{2+} quantification on robust parameters and therefore did not include analysis of first responder cells. This decision was based on the limitations of GCaMP3 kinetics and the 1 Hz temporal resolution used in our intravital imaging setup, which may not reliably capture the precise timing needed to identify such cells with confidence.

Reviewer #5 (Remarks to the Author):

We thank Reviewer #5 for his/her valuable scientific input and contribution to the review process. We also acknowledge and appreciate Nature Communications' initiative to support Early Career Researchers through co-reviewing.

REVIEWER COMMENTS (2nd Revision)

Reviewer #1 (Remarks to the Author):

Line 196 – please state in the main text here when you did the permeability experiment after transplantation (we find that nascent blood vessels shortly after implantation are leaky but that improves on full maturation). This is not a criticism it's just useful for readers interested in implantation biology to have context.

Thank you for highlighting this important point regarding vascular maturation and its potential influence on permeability measurements. We fully agree that newly formed vessels can display increased leakiness shortly after implantation, which normalizes upon maturation. In our study, the permeability testing was performed **6 weeks** after transplantation. This timing is well beyond the early post-implantation phase, and according to published data (e.g., Van der Wijk et al., Expression patterns of endothelial permeability pathways in the development of the blood-retinal barrier in mice, 2019, *FASEB J.*), endothelial barrier properties reach a mature state after approximately 4 weeks. We are therefore confident that our observations were made under conditions of a functionally mature vasculature. We have now included the time specifications in the main text, figure legend, and Materials and Methods.

Line 246-8 "In the fully anesthetized state, $[Ca^{2+}]_i$ activity was strongly suppressed, characterized by more irregular and attenuated oscillations." It is very clear that the oscillations are altered in anaesthesia but they are fundamentally still measurable and (we know from others' work) still subject to measurable insulinotropic (or at least glucose responsive) modulation. Therefore I am not comfortable with the term "strongly suppressed" because it implies that there is loss of discernible or useful oscillatory data which really isn't true. I would suggest something more like: "In the fully anesthetized state, $[Ca^{2+}]_i$ activity was significantly altered, characterized by more irregular and attenuated oscillations."

We appreciate the reviewer's careful observation. We agree that the term "strongly suppressed" may overstate the effect and could be misinterpreted as indicating a complete loss of Ca^{2+} oscillatory activity, which our study, along with others, has demonstrated is not the case. To better reflect the data, we have revised the sentence, in line with the reviewer's suggestion, to read:

"In the fully anesthetized state, $[\text{Ca}^{2+}]_i$ activity was significantly altered, characterized by more irregular and attenuated oscillations."

The discussion around slow oscillations and the potential for isoflurane to have partially uncoupled calcium dynamics from insulin secretion, with reference to Satin and others, is much improved. I don't think that basal insulin secretion is completely inhibited under isoflurane as the glucose only rises to 12 mM and insulin levels are still detectable. I do agree that this has uncovered evidence that there are probably multiple/redundant metabolic regulators of the slow oscillations (and some are clearly retained under isoflurane).

Thank you very much for the positive feedback. Indeed, insulin secretion does not appear to be completely abolished under isoflurane, just as Ca^{2+} oscillations are not entirely lost. However, the excitability of the β -cells, the translation of elevated glucose levels into an appropriate increase in insulin secretion, is clearly impaired, as demonstrated in the glucose tolerance test. Mechanistically, we do not yet have a definitive explanation, but we suspect the effect might be multifactorial.

Once again congratulations on this excellent work.

We sincerely thank the reviewer for the constructive and expert advice throughout the revision process, particularly regarding Ca^{2+} signaling, which greatly aided us in refining and strengthening key aspects of our work and manuscript.

Reviewer #2 (Remarks to the Author):

Whilst the authors have addressed most of our concerns, there are still some issues we believe refrain the manuscript from being published at this current stage.

We are grateful that the reviewer acknowledges most of their previous concerns have been satisfactorily addressed. We recognize that every study inevitably leaves open questions and opportunities for future investigation, and in our first revision we have specifically addressed key aspects such as immune status, vascular function, and graft function in terms of insulin secretion. With regard to the remaining points raised, we would like to clarify our position. In particular, concerning the question of immune integrity of the brain following graft integration within the cranial window on the dura mater, we respectfully hold a different view, which we explain in detail below.

1) "Non-invasive" is still mentioned at least thrice throughout the text.

We thank the reviewer for raising this point. While the initial cranial window surgery is inherently invasive, subsequent intravital microscopy through the pre-existing window does not require further tissue disruption. In the literature, this distinction is often reflected using terms such as "longitudinal" or "chronic" imaging, and sometimes "non-invasive" to describe the imaging sessions themselves. Regarding the reviewer's concerns, we now use "minimally invasive" throughout to accurately convey that, although the surgery is required initially, repeated imaging is performed without additional tissue trauma, making this terminology both precise and appropriate.

2) Systemic inflammation markers are not indicative of brain inflammation. Brain tissue homogenate should be analysed for inflammatory markers (e.g TNF α , IL-6, IL1 β , IFN γ), especially to reflect their claim that the craniotomy conducted does not impact on the brain. Furthermore, brain-resident cell composition and their cytokine production profile should be shown in the regions surrounding the islets vs contralateral control (microglia, astrocytes, neuron).

We appreciate this important point. In our model, however, transplanted islets are placed on the dura mater, which lies outside the brain parenchyma and is anatomically separated from it by the meninges (dura mater, subarachnoid space, and pia mater). As illustrated in Figure 1A, there is no direct contact between the transplant and brain tissue. Access to cerebrospinal fluid occurs only transiently during the initial minimal puncture of the dura, and brain tissue itself remains intact.

Because the transplantation site lies outside the blood–brain barrier, it is immunologically connected to the peripheral vascular system. We confirmed this by demonstrating high permeability of transplant vasculature (Figure 3A–D). Thus, systemic inflammation markers are appropriate to reflect immune status at this transplantation site. Importantly, a 10 kDa glucose-conjugated tracer would not leak from vessels within the brain parenchyma, since blood–brain barrier capillaries lack fenestrations and are sealed by tight junctions. In contrast, peripheral vessels readily allow diffusion of such molecules through their fenestrations and higher transcytosis activity, explaining the clear extravasation we observed in the graft vasculature.

Regarding craniotomy: our manuscript already cites a protocol study (Ref. 23) showing that cranial window surgery does not alter brain immune status when performed correctly. This technique has been widely used in hundreds of peer-reviewed studies to image cortical structures without evidence of adverse effects on brain health.

Finally, the reviewer suggests analysis of brain-resident cells (microglia, astrocytes, and neurons). Since these cells are confined to the parenchyma beneath the pia, they are not in direct contact with the graft. For this reason, analysis of brain homogenates would not be anatomically relevant to our transplantation model.

3) If direct functional proof of neuronal connectivity is beyond the scope of the present study, please do not overstate there is evidence of "innervation of islets".

We thank the reviewer for this important comment and fully agree that functional proof of neuronal connectivity is beyond the scope of the present study. We have already

clarified this in the discussion, explicitly stating that the functional activity of the nervous structures within the transplanted tissue remains to be demonstrated.

In our manuscript, the term "*innervation*" is used in its strict anatomical sense, referring only to the morphological presence and distribution of nerve fibers within the tissue. Our evidence is based on immunohistochemical analysis, which demonstrates axonal structures in the grafts. We do not intend to imply functional activity, and we have reviewed the text again to ensure this distinction is clear.

4) Our concern with transplantation of islets to the brain is not on strain difference of animals (i.e. allogeneic transplantation), but whether the introduction of islets which would be seen as foreign in the brain/dura will induce an inflammatory response and cause an recruitment of immune cells. This relates to our criticism in Point #2 above.

We agree that immune cell recruitment is a normal feature of transplantation. Transient infiltration of macrophages, neutrophils, and T cells supports debris clearance, vascularization, and tissue remodeling. Importantly, in our study this response did not progress to chronic inflammation or graft rejection, as reflected by stable systemic inflammation markers and durable graft function.

It is also well established that pancreatic islets engraft successfully at several peripheral sites, including the liver, subcutaneous tissue, intraocular space, kidney capsule, and even the subarachnoid space, without pathological immune recruitment (Refs. 28–31).

Our findings at the dura mater are consistent with these established paradigms of syngeneic transplantation.

Taken together, our data indicate that dura mater-based transplantation does not compromise immune integrity. The anatomical separation from brain parenchyma, the peripheral location outside the blood–brain barrier, the syngeneic donor–recipient setting, and the absence of chronic inflammation or graft rejection all support this conclusion. We therefore believe additional analyses of brain tissue homogenates or resident CNS cell populations are not warranted within the scope of this study.

Reviewer #3 (Remarks to the Author):

Thank you for your time, effort, and valuable scientific contribution. We also greatly appreciate the Nature Communications initiative that includes Early Career Researchers in the peer review process.

Reviewer #4 (Remarks to the Author):

We thank the authors for their thoughtful and comprehensive revisions. The updated manuscript is significantly strengthened by the inclusion of new data on vascular integration, metabolic functionality, and a more balanced discussion of both the advantages and limitations of the proposed technique. The authors have effectively addressed our main concerns and clarified the physiological relevance and translational potential of their approach. The addition of human islet transplantation data and glucose-stimulated functional responses meaningfully expands the scope and impact of the work. We are pleased to support the publication of this manuscript. As a final suggestion, we encourage the authors to adopt more precise terminology when describing their method.

We sincerely thank the reviewer for the constructive scientific input and helpful guidance throughout the revision process, which greatly assisted us in refining the manuscript—for example, by including direct evidence of insulin secretion from the transplants. In response to the final suggestion, we have carefully revisited our methods and hope that the revisions now fully address the reviewer's expectations.

While the term "non-invasive" has been removed from the title, referring to the technique throughout the manuscript as "less invasive" or "minimally invasive" would more accurately reflect the procedural refinement in comparison to other approaches involving deep tissue access or implantable devices.

We thank the reviewer for this suggestion. We agree that "minimally invasive" more accurately reflects our approach, as the initial cranial window surgery is inherently invasive, while the subsequent repeated intravital microscopy through the existing window avoids further surgical trauma. Accordingly, we have updated the manuscript to use "minimally invasive" throughout.

This is an elegant and technically impressive study that will be of broad interest to

those working in longitudinal imaging, transplantation models, and β -cell physiology. We look forward to its publication.

We sincerely thank the reviewer for this kind and encouraging words; we greatly appreciate their recognition of our work.

Reviewer #5 (Remarks to the Author):

We are grateful for the time, effort, and valuable scientific support provided. We also greatly appreciate the Nature Communications initiative that engages Early Career Researchers in the peer review process.